# M³Ret: Unleashing Zero-shot Multimodal Medical Image Retrieval via Self-Supervision

**Che Liu**[*,1,2]**, Zheng Jiang**[*,1,3]**, Chengyu Fang**[*,1,3]**, Heng Guo**[1,4]**, Yan-Jie Zhou**[1,4]**, Jiaqi Qu**[†,1,4,5]**,
Le Lu**[1]**, Minfeng Xu**[1,4]

[1]DAMO Academy, Alibaba Group    [2]Imperial College London
[3]Tsinghua University    [4]Hupan Lab    [5]Zhejiang University
[*]Equal contribution, [†]Corresponding author
**Correspondence:** {gh205191, eric.xmf, qujiaqi.qjq}@alibaba-inc.com
**Reviewed on OpenReview:** https://openreview.net/forum?id=VgmIrgbzkX

## Abstract

Medical image retrieval is essential for clinical decision-making and translational research, relying on discriminative visual representations. Yet, current methods remain fragmented, relying on separate architectures and training strategies for 2D, 3D, and video-based medical data. This modality-specific design hampers scalability and inhibits the development of unified representations. To enable unified learning, we curate a large-scale hybrid-modality dataset comprising 867,653 medical imaging samples, including 2D X-rays and ultrasounds, RGB endoscopy videos, and 3D CT scans. Leveraging this dataset, we train M³Ret, a single ViT-B-scale visual encoder without modality-specific encoders, objectives, or retrieval heads. It successfully learns transferable retrieval representations using both generative (MAE) and contrastive (SimDINO) self-supervised learning (SSL) paradigms. Our approach achieves strong and competitive zero-shot medical image retrieval across heterogeneous modalities, outperforming or matching strong baselines in several settings while specialized or larger baselines remain stronger in some cases. More remarkably, anatomy-level cross-modal retrieval emerges without paired data, and the model generalizes to unseen MRI tasks, despite never observing MRI during pretraining. Comprehensive analyses further suggest that performance improves with model and data scale. These findings deliver a promising signal to the medical imaging community, positioning M³Ret as a step toward unified visual SSL for multimodal medical image retrieval.

## 1 Introduction

In medical imaging, retrieving similar images given a query image plays a vital role in supporting clinical decision-making and advancing translational research Müller et al. (2004). Such tasks depend heavily on high-level and discriminative visual representations Datta et al. (2008); Choe et al. (2022); Qayyum et al. (2017); Li et al. (2018). However, most existing approaches focus on 2D images and rely extensively on language supervision Zhang et al. (2023); Pelka et al. (2018); Lozano et al. (2025), limiting their applicability to image-only datasets and reducing generalizability. Even self-supervised learning (SSL) methods, which avoid textual labels, often adopt modality-specific designs, training separate models for 2D and 3D data with architectures and training strategies tailored to each P'erez-Garc'ia et al. (2024); Tang et al. (2021); Wu et al. (2024a); Haghighi et al. (2022). For example, 3D models typically operate on cropped sub-volumes instead of full scans, hindering volume-level retrieval that relies on global context. This fragmentation impairs the development of unified and transferable representations, making retrieval across imaging modalities, for instance, between a 2D image and a 3D volume, practically infeasible. This challenge arises from the inherent heterogeneity of medical imaging: 2D and 3D scans differ in spatial dimensionality; grayscale modalities such as X-rays contrast sharply with RGB video (e.g., endoscopy); and temporal dynamics in videos are absent from static images. These disparities prompt a fundamental question:

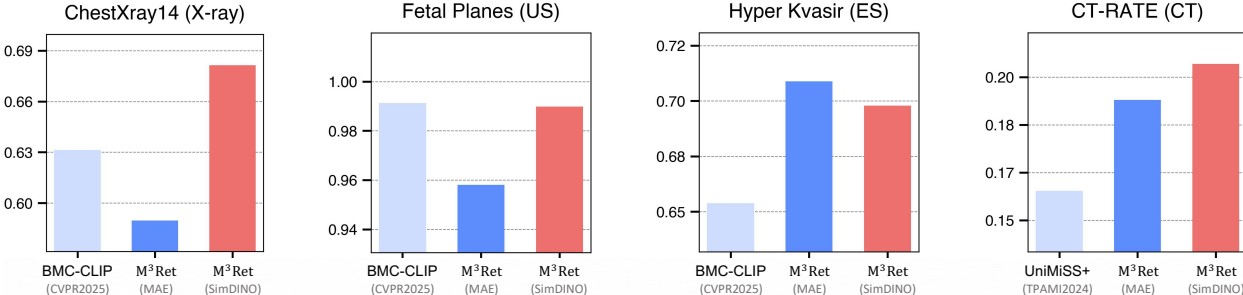

Figure 1: **Recall@5 on zero-shot image-to-image retrieval across four medical datasets.** The proposed $M^3$Ret, pretrained respectively with generative (MAE He et al. (2022)) and contrastive (SimDINO Wu et al. (2025)) self-supervision, achieves strong or competitive performance on ChestXray14 (X-ray) Wang et al. (2017), Fetal Planes (Ultrasound) Burgos-Artizzu et al. (2020), Hyper Kvasir (Endoscopy) Borgli et al. (2020), and CT-RATE (CT) Hamamci et al. (2024), compared with language-supervised (BMC-CLIP Lozano et al. (2025)) and visual SSL (UniMiSS+ Xie et al. (2024)) baselines.

> *Can we learn unified retrieval representations without modality-specific encoders, objectives, or retrieval heads?*

To address this, we propose $M^3$Ret, a unified framework for visual representation learning in zero-shot multimodal medical image retrieval. To enable generalizable pretraining, we collect a large-scale hybrid modality dataset comprising 867,653 clinical imaging samples, including 2D X-rays and ultrasounds, RGB endoscopy videos, and 3D CT scans. To the best of our knowledge, this is the largest real-world hybrid medical imaging dataset to date. Built upon this dataset, $M^3$Ret enables joint training across diverse imaging modalities without modality-specific encoders, objectives, or retrieval heads. It leverages two widely adopted SSL paradigms, MAE He et al. (2022) and SimDINO Wu et al. (2025), to learn transferable visual representations for retrieval across modalities.

We make three key contributions: **(1)** We show that diverse medical modalities, X-rays, CT scans, ultrasounds, and endoscopy videos, can be trained together in a single unified framework without modality-specific encoders, objectives, or retrieval heads. **(2)** We achieve strong and competitive performance on zero-shot image-to-image retrieval across a broad range of datasets, covering diverse imaging modalities and tasks, including global categories (various diseases or anatomical regions), region-specific abnormalities, and anatomy-level cross-modal retrieval, compared with SSL methods and supervised models with language, mask, or category supervision. Fig. 1 provides a representative illustration of the results. **(3)** We conduct an in-depth analysis of key SSL design factors, providing initial evidence that retrieval performance improves with data scale, model capacity, and compute, echoing the "bitter lesson" Sutton (2019). Our results suggest that unified visual SSL across medical imaging modalities is feasible and effective for zero-shot medical image retrieval, providing a step toward more general medical retrieval systems.

## 2 Related Work

**Medical Image Retrieval.** Medical image retrieval focuses on retrieving the most relevant medical images given a query input, facilitating tasks such as case comparison, lesion matching, and diagnostic support Müller et al. (2004); Cai et al. (2008); Kumar et al. (2013). Early approaches primarily targeted 2D images with text-based queries Zhang et al. (2023); Pelka et al. (2018); Rückert et al. (2024), while image-to-image retrieval remains relatively underexplored. Some studies Jush et al. (2023); Lee et al. (2023); Hu et al. (2022) utilize small, narrowly scoped datasets, such as binary COVID-19 classification (e.g., with or without infection), which limits both generalizability and scalability. Other modalities, like sketch-based retrieval Kobayashi et al. (2024), attempt to combine visual and textual cues, but often suffer from limited clinical usability due to complex input requirements. In 3D medical imaging, BIMCV-R Chen et al. (2024) is among the first to address retrieval but relies solely on text queries, making it incompatible with image-only datasets. 3D-MIR Abacha et al. (2023) uses image-based queries, but its slice-wise processing fails to capture full 3D

spatial context. Moreover, it supports only global category-level retrieval and does not consider fine-grained, region-level semantics. Importantly, all aforementioned methods rely on supervised learning with paired training data, which poses significant scalability challenges due to the cost and effort of annotation.

**Self-Supervised Learning for Medical Imaging.** Self-supervised learning (SSL) has emerged as a promising alternative to supervised training in the medical domain, enabling the learning of visual representations without annotation. Many existing methods P'erez-Garc'ia et al. (2024); Azizi et al. (2021); Taher et al. (2024) adopt visual-only SSL approaches, but are typically tailored to specific modalities, such as 2D chest X-rays. This limits their ability to generalize to other imaging types or anatomical regions, such as abdominal or brain scans, and to 3D modalities like CT or MRI. For 3D medical imaging, most SSL frameworks either process cropped sub-volumes Tang et al. (2021); Wu et al. (2024a); Tao et al. (2020); Zhou et al. (2021); Jiang et al. (2023); Wu et al. (2024b); Zhou et al. (2023); Xie et al. (2021), which restrict the model's capacity to learn global anatomical context, or use heavily downsampled full volumes Blankemeier et al. (2024), which may lose subtle tissue patterns. These design choices hinder the learning of comprehensive and discriminative features across the entire 3D volume.

**Multimodal Representation Learning in Medical Imaging.** Multimodal learning in medical imaging has primarily focused on aligning medical images with associated textual data, such as radiology reports Zhang et al. (2023); Pelka et al. (2018); Rückert et al. (2024); Lin et al. (2023); Hamamci et al. (2024); Zhang et al. (2025). Models like BMC-CLIP Lozano et al. (2025) leverage large-scale image-text pairs and achieve strong performance in tasks such as retrieval. However, their dependence on text supervision limits their use in scenarios where textual annotations are unavailable. Another line of work explores multimodal learning across different variants of a single imaging modality Rui et al. (2024), such as T1-weighted, T2-weighted, and FLAIR sequences in brain MRI. These approaches still operate within a single modality (e.g., MRI) and typically require data from the same anatomical region, which restricts generalization to other body parts and imaging modalities. Only a few recent efforts aim to learn representations across heterogeneous imaging modalities. For instance, UniMiSS Xie et al. (2021) and UniMiSS+ Xie et al. (2024) propose cross-modal learning between 2D X-rays and 3D CT scans. Yet, they rely on strictly aligned scans from the same anatomical region and use complex alternating training strategies, which limit scalability to more diverse and unpaired datasets. MedCoSS Ye et al. (2024) proposes a continual SSL framework to perform multimodal representation learning, but requires a multi-stage sequential training pipeline that involves clinical reports. Similarly, UniMedI He et al. (2023) aligns 2D and 3D modalities using additional report supervision, but it depends on paired image-text data, further constraining its applicability in broader clinical contexts.

# 3 Methodology

In this section, we present our unified pretraining framework M$^3$Ret, which enables visual SSL across diverse medical imaging modalities. As shown in Fig. 2, existing methods either rely on language supervision or are restricted to single modalities with limited global context. In contrast, our framework enables joint training on 2D, 3D, and video data using purely visual signals, without modality-specific encoders, objectives, retrieval heads, or additional supervision. We first describe the unified patchification that supports heterogeneous medical inputs in a modality-agnostic manner, and then introduce two SSL paradigms, MAE He et al. (2022) and SimDINO Wu et al. (2025), used for visual representation learning.

## 3.1 Unified Patchification for Multimodal Medical Images

Our framework aims to standardize the input format across various medical imaging modalities, including 2D and 3D grayscale and RGB images, to enable the use of a shared encoder. Each input image or volume is represented as a 4D tensor $X \in \mathbb{R}^{C \times H \times W \times S}$, where $C$ denotes the number of channels, $H$ and $W$ represent the height and width, which are fixed at 256 pixels, and $S$ corresponds to the number of slices in CT scans or the number of frames in endoscopy videos. We define the preprocessing and reshaping operations for each modality as follows:

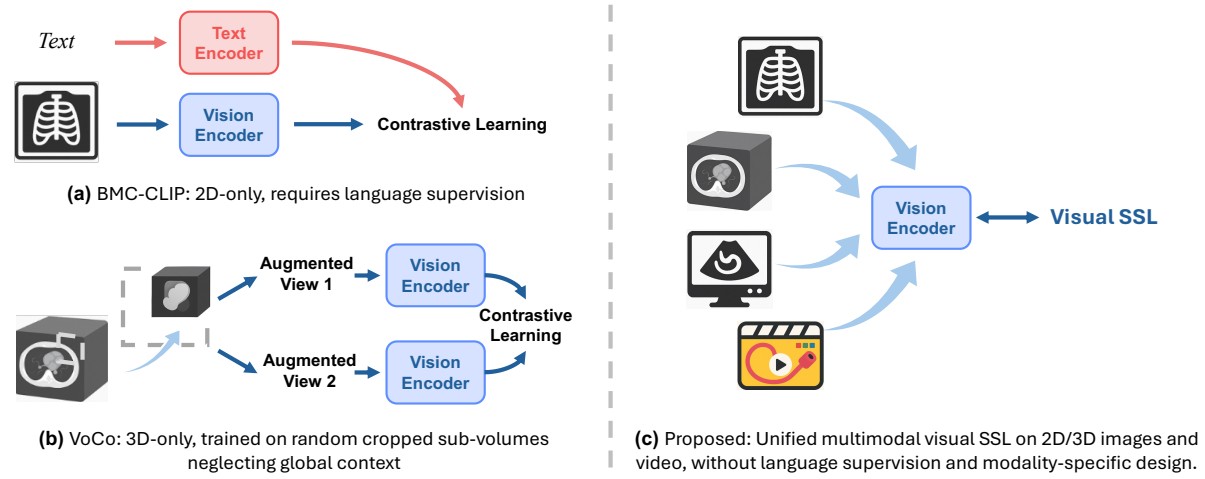

Figure 2: Comparison of visual representation learning strategies in medical imaging. **(a)** BMC-CLIP Lozano et al. (2025) relies on 2D image-text pairs (e.g., X-rays and reports), limiting generalization beyond paired data and modalities. **(b)** VoCo Wu et al. (2024b) learns from 3D sub-volumes with random cropping via contrastive learning, but neglects global context. **(c)** Our proposed approach learns unified visual representations from heterogeneous medical imaging modalities (2D, 3D, and videos) using purely visual SSL (e.g., MAE He et al. (2022), SimDINO Wu et al. (2025)), without extra supervision signals such as text or modality-specific encoders, objectives, or retrieval heads.

**X-ray and Ultrasound**: Each 2D grayscale image is resized to $256 \times 256$, duplicated along the channel dimension to match the RGB format, and replicated along the slice dimension to obtain a 4D tensor $X \in \mathbb{R}^{3 \times 256 \times 256 \times 4}$.

**Endoscopy**: Given RGB video data, we sample $S = 16$ frames from a single video, resulting in $X \in \mathbb{R}^{3 \times 256 \times 256 \times 16}$. Frames are randomly sampled during pretraining and uniformly sampled during evaluation.

**CT**: Volumes are resized or padded to $S = 64$ slices without enforcing a common physical voxel spacing or isotropic resampling. Each grayscale scan $(C = 1)$ is duplicated along the channel axis to match RGB format, yielding $X \in \mathbb{R}^{3 \times 256 \times 256 \times 64}$.

We apply a 4D patchification with patch size $P = (C_p, H_p, W_p, S_p) = (3, 16, 16, 4)$ universally, which partitions $X$ into non-overlapping spatiotemporal patches (patching is applied over $H$, $W$, and $S$; channel dimension is fixed). The temporal/depth patch size of 4 exactly divides replicated 2D images $(S = 4)$, 16-frame videos, and 64-slice CT volumes, providing a shared tokenizer while keeping the token budget feasible. This yields a sequence of visual tokens $T = \texttt{Patchify}(X) \in \mathbb{R}^{N \times D}$, where $N$ is the number of tokens and $D$ is the patch embedding dimension.

## 3.2 Masked Autoencoder (MAE)

Masked Autoencoders (MAE) He et al. (2022) aim to reconstruct the pixel-level content of masked patches from the remaining visible ones. Let the input be a 4D tensor $X \in \mathbb{R}^{C \times H \times W \times S}$. The input $X$ is partitioned into $N$ non-overlapping visual patches using a unified patchification strategy. Each patch is flattened and linearly projected into a $D$-dimensional embedding, forming the patch embedding matrix $P \in \mathbb{R}^{N \times D}$.

A fixed mask ratio $\alpha \in (0, 1)$ (e.g., $\alpha = 0.75$) determines the fraction of patches to be masked. The remaining $(1 - \alpha)N$ visible patches are denoted as $P_{\text{vis}} \in \mathbb{R}^{(1-\alpha)N \times D}$ and are processed by an encoder function $f_\theta : \mathbb{R}^{(1-\alpha)N \times D} \to \mathbb{R}^{(1-\alpha)N \times D'}$, where $\theta$ represents the learnable parameters of the encoder and $D'$ is the output dimensionality of the latent representations.

To reconstruct the masked patches, a decoder $g_\phi$ takes the encoder output along with learned embeddings for the masked patches $P_{\text{mask}}$, and outputs the reconstructed patches in pixel space: $\hat{X}_{\text{mask}} = g_\phi(f_\theta(P_{\text{vis}}), P_{\text{mask}})$.

Here, $\phi$ denotes the learnable parameters of the decoder, and $\hat{X}_{\text{mask}}$ represents the reconstructed pixel values of the masked patches.

The model is trained to minimize the mean squared error (MSE) between the reconstructed and original pixel values of the masked patches. Let $\mathcal{M} \subset \{1, \ldots, N\}$ be the set of indices corresponding to the masked patches. The training objective is given by:

$$\mathcal{L}_{\text{MAE}} = \frac{1}{\alpha N} \sum_{i \in \mathcal{M}} \|\hat{X}_i - X_i\|_2^2,$$

where $X_i$ and $\hat{X}_i$ denote the original and reconstructed pixel values of patch $i$, respectively.

Notably, MAE does not utilize a `[CLS]` token. In downstream zero-shot image retrieval tasks, we use the average pooling of all visual patch embeddings from the encoder as the image representation to compute cosine similarity between images.

### 3.3 SimDINO

SimDINO Wu et al. (2025) aims to learn discriminative visual representations by aligning features extracted from different augmented views of the same input. Given an input image $X$, two types of augmented views are generated, local $v_c(X)$ and global $v_g(X)$, as detailed in Appendix B. We sample two global views for every modality, plus 10 local views for 2D inputs and 4 for 3D or video inputs to avoid GPU out-of-memory issues. Each view is partitioned into non-overlapping visual patches with a unified patchification strategy, and a `[CLS]` token is prepended. All views pass through the student encoder $f_{\text{student}}$, whereas only global views feed the teacher encoder $f_{\text{teacher}}$. The resulting `[CLS]` embeddings,

$$z_c = f_{\text{student}}^{\text{cls}}(v_c(X)), \quad z_g = f_{\text{teacher}}^{\text{cls}}(v_g(X)),$$

present discriminative visual representations. The total loss combines an alignment term with coding-rate regularization Ma et al. (2007); Yu et al. (2020):

$$\mathcal{L}_{\text{SimDINO}} = \frac{1}{2}\|z_c - z_g\|_2^2 - \frac{1}{2}\log\det\left(I + \frac{d}{\epsilon^2}\Gamma\right)$$

where $\Gamma = \text{Cov}[z_c] \in \mathbb{R}^{d \times d}$ denotes the covariance matrix of the student `[CLS]` embeddings in a batch, with $d$ being the embedding dimension. We set $\epsilon = 0.5$, and update the teacher parameters using exponential moving average (EMA) of the student parameters, where the momentum starts at 0.996 and is gradually increased to 1.0 using a cosine schedule during pretraining.

During zero-shot image retrieval tasks, SimDINO uses the original (non-augmented) image to compute its representation by concatenating the `[CLS]` token with the average-pooled patch embeddings. The cosine similarity between these combined representations is then used to rank image pairs.

### 3.4 Pretraining Configuration

**Dataset.** We collect a total of 867,653 samples from our collaborating hospital, spanning four imaging modalities: 2D X-ray, 2D ultrasound, endoscopy videos, and 3D CT scans. The data distribution is shown in Appendix A. Preprocessing includes: (1) resizing the longer axis of the height and width to 256 pixels and zero-padding the shorter axis to preserve aspect ratio and avoid distortion; (2) sampling 16 frames per endoscopy video; and (3) resizing or padding CT scans to 64 slices and clipping HU values to $[-1000, 1000]$. All intensity values are normalized to the range of $[0, 1]$.

**Implementation Details.** We use Vision Transformer (ViT) Dosovitskiy et al. (2021) as the vision encoder for both MAE and SimDINO, with our 4D patchification strategy described in Sec. 3.1, and follow their official implementations. The maximum learning rate is set to $1 \times 10^{-3}$, with a linear warm-up for the first 10 epochs, followed by a cosine annealing schedule. Both models are pretrained for 300 epochs over the pooled pretraining set. Since different modalities yield varying numbers of visual tokens and thus memory demands, we use local batch sizes of 16 for CT scans and 32 for other modalities, without modality-specific

oversampling. Pretraining is performed on 16 Nvidia H20 GPUs. Each GPU worker receives a same-modality local batch with a consistent shape, while the global batch can contain multiple modalities across workers. Notably, MAE uses no data augmentation beyond token masking, while SimDINO's augmentations are detailed in Appendix B.

## 4 Zero-shot Medical Image Retrieval

### 4.1 Task Configuration

To comprehensively evaluate our model's performance in zero-shot medical image-to-image retrieval, we design four distinct task settings, each targeting a specific level of granularity or imaging modality. These tasks are illustrated below and aim to assess the model's capability across various retrieval scenarios. We report Recall@K, Median Rank (MdR), and Mean Rank (MnR) as evaluation metrics. Details of the datasets used in each setting are provided in Appendix C.

**Category-level Retrieval:** This setting evaluates the model's basic capability to retrieve images based on global categories. The input query image may belong to one or multiple categories at the image level, such as specific diseases (e.g., "atelectasis") or anatomical regions (e.g., "head"). A retrieval is considered correct if the returned sample shares at least one positive category with the query image. For this task, we use the following public datasets: ChestXray14 Wang et al. (2017) for X-ray modality, Fetal Planes Burgos-Artizzu et al. (2020) for Ultrasound, and Kvasir Capsule Smedsrud et al. (2020) and Hyper Kvasir Borgli et al. (2020) for Endoscopy.

**Progressive Regional Retrieval Tasks:** To evaluate the model's understanding of fine-grained regional information, we design two increasingly challenging tasks using the RadGenome-ChestCT Zhang et al. (2024) dataset, which is extended from CT-RATE Hamamci et al. (2024) dataset with anatomical region grounded reports (We use the term "CT-RATE" consistently throughout the descriptions for its simplicity). These tasks assess performance at different levels of spatial granularity: **(1) Regional Abnormality Retrieval:** This task evaluates the model's ability to retrieve images based on coarse regional annotations. Each query image is labeled with regional abnormality statuses (e.g., the "aorta" labeled as "normal" or "abnormal"). A retrieval is considered correct if the returned image matches the query's abnormality status for the corresponding regions. **(2) Lesion Size Retrieval:** This more challenging task requires retrieval based on precise lesion size annotations. Each query image is labeled with regional lesion descriptions (e.g., "abdomen" with "hypodense lesion, 20 mm"). A retrieval is correct if the returned image matches the lesion size annotations for the relevant regions. Together, these tasks assess the model's ability to capture increasingly fine-grained pathological information, from coarse abnormality status to precise lesion characterization.

**Anatomy-level Cross-modal Retrieval:** This setting evaluates the model's ability to generalize across imaging modalities, including those not seen during pretraining. Given a query image, the task is to retrieve an image from a different modality that shares the same mapped anatomical region or body part. For example, given an X-ray image of the "head", the desired output is a corresponding head CT scan.

In addition to cross-modal retrieval between modalities seen during pretraining (e.g., X-ray and CT), we also explore a more challenging scenario: retrieving MRI images based on CT queries. This task is particularly demanding because the model has never been exposed to MRI data during pretraining, making it a true test of generalization to unseen modalities. For instance, given a CT image of the "abdomen", the desired output is a corresponding abdomen MRI.

These tasks probe anatomy-level cross-modal organization in the frozen visual representation, rather than pathology-level or same-patient retrieval.

### 4.2 Baselines

We select strong baseline methods that cover diverse supervision learning strategies, including text, mask, and disease category supervision, as well as visual-only SSL, to enable comprehensive comparisons. These baselines include models pretrained with 2D images, 3D volumes, and joint 2D-3D inputs. For all experiments,

we use their official pretrained weights and inference code. Additional modality-specific specialist baselines, including FetalCLIP for ultrasound and EndoViT for endoscopy, are reported in Appendix D.

**DINOv2 Oquab et al. (2023):** DINOv2 is a self-supervised learning model that learns powerful visual features from a large, curated dataset of 142 million images without relying on textual annotations. It employs self-distillation and a ViT architecture to learn versatile visual features.

**SigLIP2 Tschannen et al. (2025):** SigLIP2 is a multilingual vision-language model that learns generalizable visual representations from image-text pairs. It is trained on the extensive web-crawled dataset, which contains 10 billion images with their corresponding captions in over 100 languages. The training process combines a sigmoid loss for image-text matching with self-supervised techniques, including self-distillation and masked prediction, to enhance performance on both global and dense prediction tasks.

**DINOv3 Siméoni et al. (2025):** DINOv3 is a self-supervised learning model that significantly scales up its predecessor, DINOv2, by training on a massive dataset of 1.7 billion images with various scale ViT architecture.

**BMC-CLIP Lozano et al. (2025):** BMC-CLIP is trained on a 24M image-text pair dataset (BIOMED-ICA), restricted to 2D image inputs and reliant solely on text supervision. Its reliance on large-scale text-image data collection incurs significant costs and limits scalability.

**VoCo Wu et al. (2024b):** VoCo is pretrained on 160K CT scans to learn volumetric representations but uses cropped sub-volumes as input rather than full 3D volumes, which limits its ability to capture global context. It supports only 3D CT inputs, making it unsuitable for 2D-based tasks or cross-modal retrieval. Since VoCo provides multiple official pretrained weights[1], we select the model with a comparable scale to ours, VoCo-Omni-B, as the baseline. Notably, this version of VoCo is pretrained using both mask annotations and a segmentation task in parallel with contrastive learning.

**Merlin Blankemeier et al. (2024):** Merlin is pretrained on over 6 million CT-EHR pairs with full supervision, using EHR codes as disease categories, and is restricted to 3D CT inputs. Its reliance on paired CT-EHR data limits its applicability to other modalities or datasets lacking such annotations.

**CT-FM Pai et al. (2025):** CT-FM is a large-scale, 3D visual foundation model pre-trained on 148,000 3D CT scans using contrastive learning in subvolume-level. The model is designed to perform diverse and complex tasks across imaging modalities, including segmentation, triage, retrieval, and semantic understanding.

**CT-CLIP Hamamci et al. (2024):** CT-CLIP is a contrastive language-image pre-training framework focused on 3D medical imaging, specifically CT scans. It is trained on the CT-RATE dataset, which pairs 25,692 non-contrast 3D chest CT scans with their corresponding radiology reports. The model learns to align 3D visual features with textual representations from the reports, enabling it to understand the semantic content of medical images. This allows CT-CLIP to be applied to various downstream tasks, such as multi-abnormality detection and case retrieval, without requiring task-specific training.

**UniMiSS+ Xie et al. (2024):** UniMiSS+ is pretrained in a purely self-supervised manner using Chest X-ray and CT data. It aims to learn 2D and 3D representations, but the pretraining explicitly aligns 2D and 3D scans from the same body part, hence it is limited to cases where both modalities originate from the same anatomical region.

### 4.3 Experimental Results

**Zero-shot retrieval across modalities.** We evaluate $M^3$Ret on four datasets covering different modalities, ChestXray14 (X-ray), Fetal Planes (ultrasound), Kvasir Capsule and Hyper Kvasir (endoscopy), under zero-shot image-to-image retrieval settings. As shown in Table 1, $M^3$Ret with SimDINO pretraining achieves strong and competitive performance against BMC-CLIP Lozano et al. (2025), a strong baseline pretrained on 24M image-text pairs with language supervision, and UniMiSS+ Xie et al. (2024), a domain-specific SSL method trained on chest X-rays and CT scans. It is strongest on ChestXray14 and Hyper Kvasir, while larger or modality-specific baselines remain stronger in some settings. Results with overlapping confidence

---

[1]https://github.com/Luffy03/Large-Scale-Medical

Table 1: Comparison of category-level retrieval. The best and second-best results are highlighted in **bold** and underlined. † indicates models using language supervision. For compactness, 95% bootstrap CIs are shown for the key R@1/R@5 metrics of M³Ret; full CIs for all methods and metrics are reported in Appendix D.

| Modality | X-ray | | | | | Ultrasound | | | | |
|---|---|---|---|---|---|---|---|---|---|---|
| Dataset | ChestXray14 | | | | | Fetal Planes | | | | |
| Metric | R@1 ↑ | R@5 ↑ | R@10 ↑ | MnR ↓ | MdR ↓ | R@1 ↑ | R@5 ↑ | R@10 ↑ | MnR ↓ | MdR ↓ |
| DINOv2-B Oquab et al. (2023) | 0.311 | 0.658 | 0.798 | 6.277 | 6.246 | 0.890 | 0.982 | 0.992 | 5.462 | 5.444 |
| DINOv3-7B Siméoni et al. (2025) | 0.304 | 0.649 | 0.793 | 6.277 | 6.250 | 0.939 | 0.994 | 0.997 | 5.482 | 5.474 |
| SigLIP2† Tschannen et al. (2025) | 0.247 | 0.599 | 0.760 | 6.636 | 6.631 | 0.902 | 0.987 | 0.995 | 5.465 | 5.449 |
| BMC-CLIP† Lozano et al. (2025) | 0.284 | 0.631 | 0.777 | 6.479 | 6.460 | 0.947 | 0.991 | 0.996 | 5.484 | 5.474 |
| UniMiSS+ Xie et al. (2024) | 0.298 | 0.648 | 0.783 | 6.384 | 6.354 | / | / | / | / | / |
| M³Ret (MAE) | 0.247 ±0.010 | 0.603 ±0.011 | 0.762 | 6.575 | 6.552 | 0.881 ±0.008 | 0.966 ±0.005 | 0.981 | 5.429 | 5.399 |
| M³Ret (SimDINO) | 0.345 ±0.010 | 0.674 ±0.010 | 0.812 | 6.136 | 6.095 | 0.955 ±0.006 | 0.990 ±0.002 | 0.996 | 5.456 | 5.445 |

| Modality | Endoscopy | | | | | | | | | |
|---|---|---|---|---|---|---|---|---|---|---|
| Dataset | Kvasir Capsule | | | | | Hyper Kvasir | | | | |
| Metric | R@1 ↑ | R@5 ↑ | R@10 ↑ | MnR ↓ | MdR ↓ | R@1 ↑ | R@5 ↑ | R@10 ↑ | MnR ↓ | MdR ↓ |
| DINOv2-B Oquab et al. (2023) | 0.784 | 0.898 | 0.936 | 5.132 | 5.038 | 0.396 | 0.668 | 0.736 | 6.788 | 6.801 |
| DINOv3-7B Siméoni et al. (2025) | 0.776 | 0.913 | 0.939 | 5.191 | 5.106 | 0.387 | 0.673 | 0.731 | 6.850 | 6.865 |
| SigLIP2† Tschannen et al. (2025) | 0.720 | 0.895 | 0.927 | 5.245 | 5.134 | 0.374 | 0.657 | 0.731 | 6.775 | 6.755 |
| BMC-CLIP† Lozano et al. (2025) | 0.743 | 0.904 | 0.936 | 5.186 | 5.117 | 0.390 | 0.654 | 0.706 | 6.842 | 6.817 |
| M³Ret (MAE) | 0.606 ±0.050 | 0.843 ±0.039 | 0.901 | 5.360 | 5.289 | 0.412 ±0.048 | 0.687 ±0.049 | 0.753 | 6.592 | 6.563 |
| M³Ret (SimDINO) | 0.647 ±0.051 | 0.872 ±0.038 | 0.913 | 5.302 | 5.201 | 0.448 ±0.048 | 0.690 ±0.049 | 0.747 | 6.617 | 6.657 |

Table 2: Comparison of retrieval at the levels of regional abnormality status and lesion-size buckets. ‡ and § indicate supervised pretraining with segmentation masks, and disease categories, respectively. For compactness, 95% bootstrap CIs are shown for the key R@1/R@5 metrics of M³Ret; full CIs are reported in Appendix D.

| Method | Regional Abnormality | | | | | Lesion Size | | | | |
|---|---|---|---|---|---|---|---|---|---|---|
| | R@1 ↑ | R@5 ↑ | R@10 ↑ | MnR ↓ | MdR ↓ | R@1 ↑ | R@5 ↑ | R@10 ↑ | MnR ↓ | MdR ↓ |
| CT-FM Pai et al. (2025) | 0.044 | 0.173 | 0.273 | 9.493 | 9.495 | 0.007 | 0.031 | 0.059 | 10.666 | 10.666 |
| CT-CLIP Hamamci et al. (2024) | 0.038 | 0.151 | 0.243 | 9.649 | 9.650 | 0.004 | 0.022 | 0.045 | 10.760 | 10.760 |
| VoCo‡ Wu et al. (2024a) | 0.037 | 0.150 | 0.241 | 9.662 | 9.662 | 0.004 | 0.021 | 0.043 | 10.773 | 10.773 |
| UniMiSS+ Xie et al. (2024) | 0.040 | 0.159 | 0.255 | 9.591 | 9.593 | 0.006 | 0.028 | 0.050 | 10.711 | 10.711 |
| Merlin§ Blankemeier et al. (2024) | 0.054 | 0.198 | 0.303 | 9.311 | 9.312 | 0.010 | 0.043 | 0.076 | 10.571 | 10.572 |
| M³Ret (MAE) | 0.050 ±0.003 | 0.188 ±0.005 | 0.295 | 9.378 | 9.376 | 0.012 ±0.003 | 0.037 ±0.005 | 0.070 | 10.598 | 10.597 |
| M³Ret (SimDINO) | 0.058 ±0.003 | 0.200 ±0.005 | 0.303 | 9.298 | 9.299 | 0.014 ±0.003 | 0.046 ±0.005 | 0.076 | 10.544 | 10.544 |

intervals or very small margins are treated as comparable rather than clear improvements. Furthermore, M³Ret with SimDINO outperforms MAE variants on 3 out of 4 datasets, suggesting it learns more discriminative and transferable retrieval features by aligning views rather than reconstructing pixels. These results highlight M³Ret's ability to extract visual representations using a unified encoder and retrieval protocol without modality-specific encoders, objectives, or retrieval heads. Fig. 5 displays top 3 retrieved examples computed by M³Ret and BMC-CLIP Lozano et al. (2025) for a qualitative comparison.

**Retrieval based on local abnormality.** We further assess M³Ret on fine-grained retrieval tasks involving regional abnormality status and lesion-size buckets (Table 2). M³Ret with SimDINO achieves competitive performance against supervised baselines such as VoCo Wu et al. (2024b), which relies on 160K pixel-level organ and tumor annotations, and Merlin, which is trained with disease-level supervision. Our approach achieves these results using purely self-supervised learning, without any task-specific design or annotations. This suggests that generic visual SSL can capture localized pathological cues useful for fine-grained retrieval without region-level supervision.

**Anatomy-level cross-modal generalization.** To explore anatomy-level cross-modal retrieval, we benchmark M³Ret on three setups: CT ↔ MRI, CT ↔ X-ray, and MRI ↔ X-ray (Table 3). A retrieved sample is counted as correct when the query and candidate map to the same anatomical region or body part. M³Ret

Table 3: Anatomy-level cross-modal retrieval performance across tasks and metrics. The best and second-best results are highlighted in **bold** and underlined. ‡ and § indicate supervised pretraining with segmentation masks, and disease categories, respectively. For compactness, 95% bootstrap CIs are shown for the key R@1/R@5 metrics of M³Ret; full CIs are reported in Appendix D.

| | CT ↔ MRI | | | | | | | | | |
|---|---|---|---|---|---|---|---|---|---|---|
| Task | CT → MRI | | | | | MRI → CT | | | | |
| Metric | R@1 ↑ | R@5 ↑ | R@10 ↑ | MnR ↓ | MdR ↓ | R@1 ↑ | R@5 ↑ | R@10 ↑ | MnR ↓ | MdR ↓ |
| UniMiSS+ | 0.160 | 0.303 | 0.370 | 9.126 | 9.142 | 0.207 | 0.315 | 0.380 | 8.802 | 8.790 |
| VoCo‡ | 0.258 | 0.561 | 0.676 | 7.116 | 7.056 | 0.196 | 0.342 | 0.424 | 8.692 | 8.677 |
| Merlin§ | 0.370 | **0.626** | **0.710** | **7.005** | **6.977** | 0.236 | 0.463 | 0.548 | 7.895 | 7.871 |
| M³Ret (MAE) | 0.301 ±0.016 | 0.525 ±0.017 | 0.641 | 7.552 | 7.554 | 0.246 ±0.011 | 0.329 ±0.011 | 0.397 | 8.661 | 8.591 |
| M³Ret (SimDINO) | **0.424** ±0.014 | 0.607 ±0.016 | 0.677 | 7.285 | 7.274 | **0.252** ±0.011 | **0.495** ±0.013 | **0.581** | **7.746** | **7.767** |

| | CT ↔ X-ray | | | | | | | | | |
|---|---|---|---|---|---|---|---|---|---|---|
| Task | CT → X-ray | | | | | X-ray → CT | | | | |
| Metric | R@1 ↑ | R@5 ↑ | R@10 ↑ | MnR ↓ | MdR ↓ | R@1 ↑ | R@5 ↑ | R@10 ↑ | MnR ↓ | MdR ↓ |
| UniMiSS+ | 0.204 | 0.336 | 0.417 | 8.778 | 8.774 | **0.344** | 0.518 | 0.595 | 7.745 | 7.720 |
| M³Ret (MAE) | **0.327** ±0.010 | **0.513** ±0.010 | **0.572** | **7.837** | **7.799** | 0.310 ±0.007 | **0.552** ±0.008 | **0.625** | **7.403** | **7.396** |
| M³Ret (SimDINO) | 0.262 ±0.012 | 0.487 ±0.014 | 0.548 | 7.912 | 7.900 | 0.285 ±0.009 | 0.502 ±0.010 | 0.581 | 7.851 | 7.836 |

| | MRI ↔ X-ray | | | | | | | | | |
|---|---|---|---|---|---|---|---|---|---|---|
| Task | MRI → X-ray | | | | | X-ray → MRI | | | | |
| Metric | R@1 ↑ | R@5 ↑ | R@10 ↑ | MnR ↓ | MdR ↓ | R@1 ↑ | R@5 ↑ | R@10 ↑ | MnR ↓ | MdR ↓ |
| UniMiSS+ | 0.012 | 0.044 | 0.073 | 10.599 | 10.597 | 0.101 | 0.153 | 0.192 | 9.895 | 9.890 |
| M³Ret (MAE) | 0.055 ±0.003 | 0.162 ±0.006 | 0.217 | 9.802 | 9.792 | 0.130 ±0.005 | 0.200 ±0.009 | 0.242 | 9.718 | 9.717 |
| M³Ret (SimDINO) | **0.152** ±0.008 | **0.306** ±0.012 | **0.373** | **8.787** | **8.798** | **0.165** ±0.008 | **0.382** ±0.011 | **0.482** | **8.383** | **8.388** |

achieves strong performance in most configurations and generalizes well to MRI-related tasks without ever seeing MRI data during pretraining. In CT ↔ X-ray retrieval, MAE outperforms SimDINO and UniMiSS+ in the CT → X-ray direction, while UniMiSS+ remains strongest for X-ray → CT R@1. We hypothesize that CT-X-ray retrieval may require matching volumetric anatomical structure to 2D projection appearance, and that MAE's reconstruction objective may preserve global projection-relevant structure; we present this as a plausible explanation rather than a causal conclusion. Overall, these findings show that a unified visual SSL framework can organize heterogeneous modalities at the anatomy level without paired samples or explicit cross-modal alignment.

### 4.4 Analysis

**Effect of the Number of Local Views.** We investigate how varying the number of local crops for 2D images affects the performance of M³Ret pretrained with SimDINO. Specifically, we evaluate settings with $\{2, 4, 6, 8, 10\}$ local views, as shown in Fig. 3 (a). Across different numbers of local crops, the performance remains stable without noticeable oscillations, suggesting that learned visual representation via SSL is not sensitive to the number of local views. Interestingly, this observation contrasts with findings from several SSL studies on natural images Caron et al. (2021; 2020), which report that increasing the number of local views enhance representation learning on RGB images. A possible explanation lies in the nature of medical images: even when pretraining across multiple imaging modalities, samples from the same modality or anatomical region often exhibit highly similar structures, differing only in subtle local patterns. As a result, additional local crops may provide limited incremental benefit for representation learning in this domain.

**Effect of the Embedding Strategy.** We further assess the robustness of our model with respect to different embedding strategies, including average pooling over all visual tokens except the `[CLS]` token (Avg.),

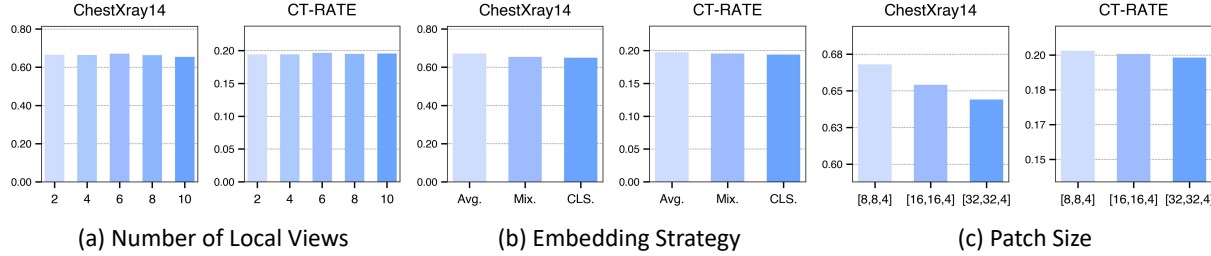

Figure 3: **Recall@5 for ChestXray14 and CT-RATE datasets. (a)** Performance under varying numbers of local crops. **(b)** Effect of embedding strategies. **(c)** Comparison of different patch sizes. $M^3$Ret (SimDINO) shows stable performance across local views and embedding choices, while smaller patch size offers better granularity and improves representation learning capacity.

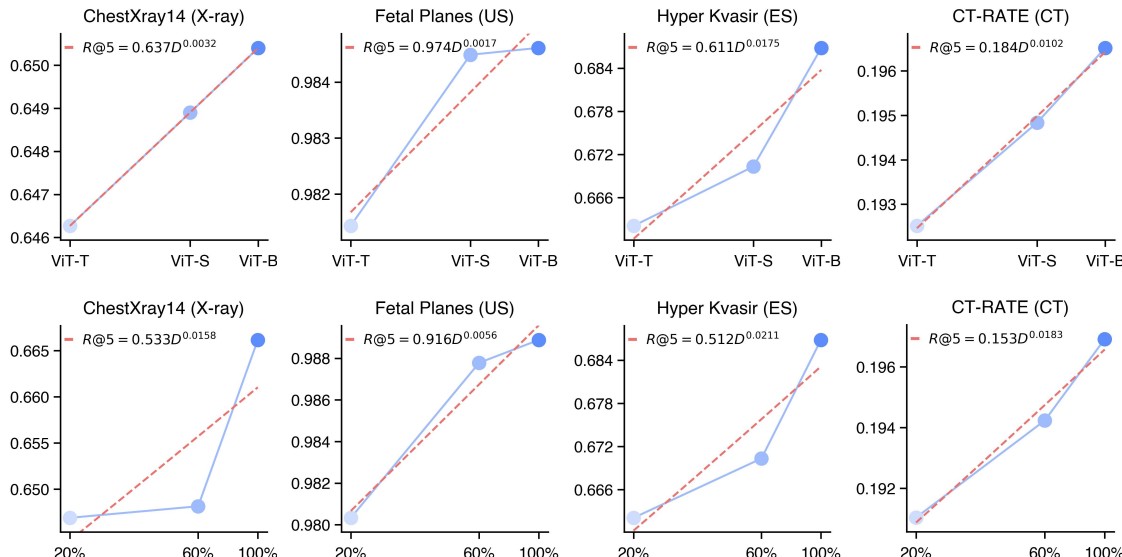

Figure 4: **Recall@5 across four downstream tasks. Top**: effect of model size using ViT-T, ViT-S, and ViT-B. **Bottom**: effect of increasing pretraining data ratio (20%, 60%, 100%). $M^3$Ret consistently benefits from larger models and more data, suggesting that performance improves with scale. Red dashed lines show preliminary fitted trends.

a mixed strategy (Mix.) that concatenates the average-pooled visual tokens with the `[CLS]` token, and using the `[CLS]` token alone. As illustrated in Fig. 3 (b), performance remains consistent across all embedding methods on both ChestXray14 and CT-RATE datasets. This stability indicates that our framework is not sensitive to the choice of embedding approach, highlighting the robustness of the learned representations.

**Effect of Patch Size.** To explore the effect of feature granularity, we pretrain $M^3$Ret using three different patch sizes: $[8 \times 8 \times 4]$, $[16 \times 16 \times 4]$, and $[32 \times 32 \times 4]$ (The first dimension is always kept as 3 and is omitted here for simplicity). This allows us to examine impact of patch granularity on representation quality. As shown in Fig. 3 (c), we observe a clear trend: smaller patch sizes consistently improve performance across various downstream tasks. This suggests that finer-grained features are more effective for representation learning in medical imaging. These results also align with findings from visual SSL on natural images Oquab et al. (2023); Caron et al. (2021), indicating that fine-grained features are beneficial across both natural and medical domains.

**Model Parameter Scaling.** We examine empirical scaling with respect to model size by comparing three ViT variants: ViT-T, ViT-S, and ViT-B. As shown in the top of Fig. 4, $M^3$Ret achieves consistent performance gains across four downstream datasets with various modalities as model capacity increases, suggesting that performance improves with model scale.

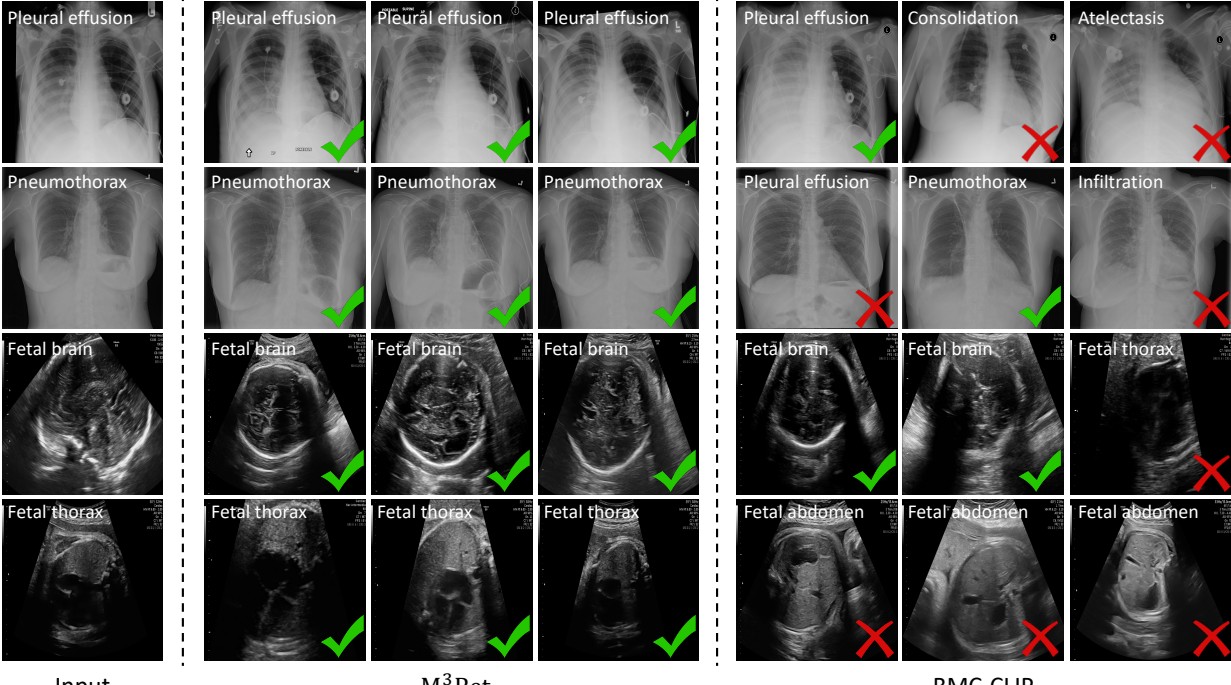

Figure 5: **Qualitative comparison of top 3 retrieval examples.** The top two rows show two X-ray query images presenting pleural effusion and pneumothorax, respectively. The bottom two rows display two ultrasound query images capturing the fetal brain and thorax, respectively. The top 3 retrieved results based on embedding similarity are listed in each row.

**Pretraining Data Scaling.** To assess empirical scaling with respect to dataset size, we pretrain on 20%, 60%, and 100% of the dataset. The bottom of Fig. 4 shows performance consistently improves, suggesting that M³Ret effectively leverages more training data. Our analysis shows preliminary scaling trends, with fitted slopes and intercepts computed following Blankemeier et al. (2024). These results provide initial evidence that retrieval performance improves with both model capacity and data size, rather than establishing a universal scaling law.

**Unimodal and Multimodal Visual Pretraining.** We analyze the impact of unimodal versus multimodal pretraining on retrieval performance across different visual modalities. Overall, multimodal pretraining tends to improve retrieval metrics across most datasets. However, there are cases where unimodal pretraining performs competitively or slightly better in certain settings. Notably, multimodal pretraining does not degrade performance significantly compared to unimodal pretraining, even when unimodal models are stronger in some settings. This suggests that multimodal pretraining can enhance retrieval representations without introducing issues such as modality collapse or conflicts during training. The results in Table 4 demonstrate that incorporating multiple modalities in visual pretraining helps the model capture diverse and complementary features without negative side effects.

## 5 Conclusion

In this work, we demonstrate that unified visual representation learning across heterogeneous modalities is feasible and effective for zero-shot medical image retrieval, without relying on modality-specific encoders, objectives, or retrieval heads. It can be accomplished through the integration of a large-scale, real-world multimodal dataset and purely visual SSL, with both SimDINO and MAE proving successful. Our unified model, M³Ret, achieves strong and competitive retrieval performance and demonstrates generalization to unseen modalities such as MRI and to fine-grained pathological cues, all without paired data, segmentation masks, or language supervision. M3Ret outperforms DINOv3-7B on ChestXray14 and HyperKvasir, while

Table 4: Ablation study results comparing unimodal and multimodal pretraining for category-level retrieval. The best results are highlighted in **bold**.

| Modality | X-ray | | | | | Ultrasound | | | | |
|---|---|---|---|---|---|---|---|---|---|---|
| Dataset | ChestXray14 | | | | | Fetal Planes | | | | |
| Metric | R@1 ↑ | R@5 ↑ | R@10 ↑ | MnR ↓ | MdR ↓ | R@1 ↑ | R@5 ↑ | R@10 ↑ | MnR ↓ | MdR ↓ |
| Unimodal | 0.295 | 0.650 | 0.785 | 6.343 | 6.314 | 0.926 | 0.980 | 0.991 | **5.449** | **5.427** |
| Multimodal | **0.323** | **0.665** | **0.795** | **6.259** | **6.220** | **0.934** | 0.980 | 0.991 | 5.468 | 5.453 |

| Modality | Endoscopy | | | | | | | | | |
|---|---|---|---|---|---|---|---|---|---|---|
| Dataset | Kvasir Capsule | | | | | Hyper Kvasir | | | | |
| Metric | R@1 ↑ | R@5 ↑ | R@10 ↑ | MnR ↓ | MdR ↓ | R@1 ↑ | R@5 ↑ | R@10 ↑ | MnR ↓ | MdR ↓ |
| Unimodal | 0.534 | 0.825 | **0.918** | **5.411** | **5.340** | 0.360 | 0.673 | **0.755** | 6.754 | 6.780 |
| Multimodal | **0.548** | **0.834** | 0.901 | 5.431 | 5.353 | **0.432** | 0.673 | 0.736 | **6.700** | **6.672** |

| Modality | CT | | | | | | | | | |
|---|---|---|---|---|---|---|---|---|---|---|
| Dataset | Regional Abnormality | | | | | Lesion Size | | | | |
| Metric | R@1 ↑ | R@5 ↑ | R@10 ↑ | MnR ↓ | MdR ↓ | R@1 ↑ | R@5 ↑ | R@10 ↑ | MnR ↓ | MdR ↓ |
| Unimodal | 0.053 | 0.194 | **0.296** | 9.344 | 9.343 | **0.015** | **0.045** | 0.075 | **10.543** | **10.543** |
| Multimodal | 0.053 | **0.195** | 0.295 | **9.340** | **9.342** | 0.013 | 0.043 | **0.076** | 10.562 | 10.562 |

specialized or larger baselines remain strongest in some modality-specific settings. Additionally, we find that M³Ret pretrained with SimDINO provides better performance than the MAE-pretrained variant in most scenarios, and provides preliminary evidence that retrieval performance improves with model and data scale. Our results provide initial evidence toward building scalable visual SSL systems for multimodal medical image retrieval.

## Broader Impact

This work delivers a key finding to the medical imaging and broader medical imaging computing communities: visual SSL can support strong zero-shot retrieval across heterogeneous medical imaging modalities without relying on modality-specific architectural design or paired supervision. By demonstrating competitive retrieval performance and anatomy-level generalization from purely visual signals, our findings highlight the potential of visual SSL as a unified approach for medical image retrieval. Our results also emphasize the critical role of data scale. We show that increasing the quantity of training data yields substantial performance gains, underscoring the need for future efforts to prioritize large-scale, high-quality medical image collection and curation. We hope this work encourages the community to build diverse multimodal datasets for learning unified visual representations that can support clinical and scientific retrieval workflows.

Medical retrieval systems also carry deployment risks. Retrieved neighbors can be over-interpreted in clinical settings, and positive/negative pair definitions may be ambiguous when cases share labels but differ in subtype, severity, location, or partially overlapping findings. The system should therefore not be used as an autonomous diagnostic tool. Practical deployment would require external and prospective validation, human-in-the-loop use, uncertainty reporting, and clear separation between research retrieval support and clinical decision-making. All private data were de-identified before model development and evaluation, and patient identifiers were not used as model inputs, retrieval keys, or similarity features.

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

# Contents

## A  Pretraining Data Distribution

In this section, we illustrate the distribution of body parts in the pretraining dataset across four modalities: X-ray, ultrasound, endoscopy, and CT, as shown in Fig. 6. For better visualization, we only include body parts with more than 1,000 samples. The overall modality distribution is summarized in Table 5.

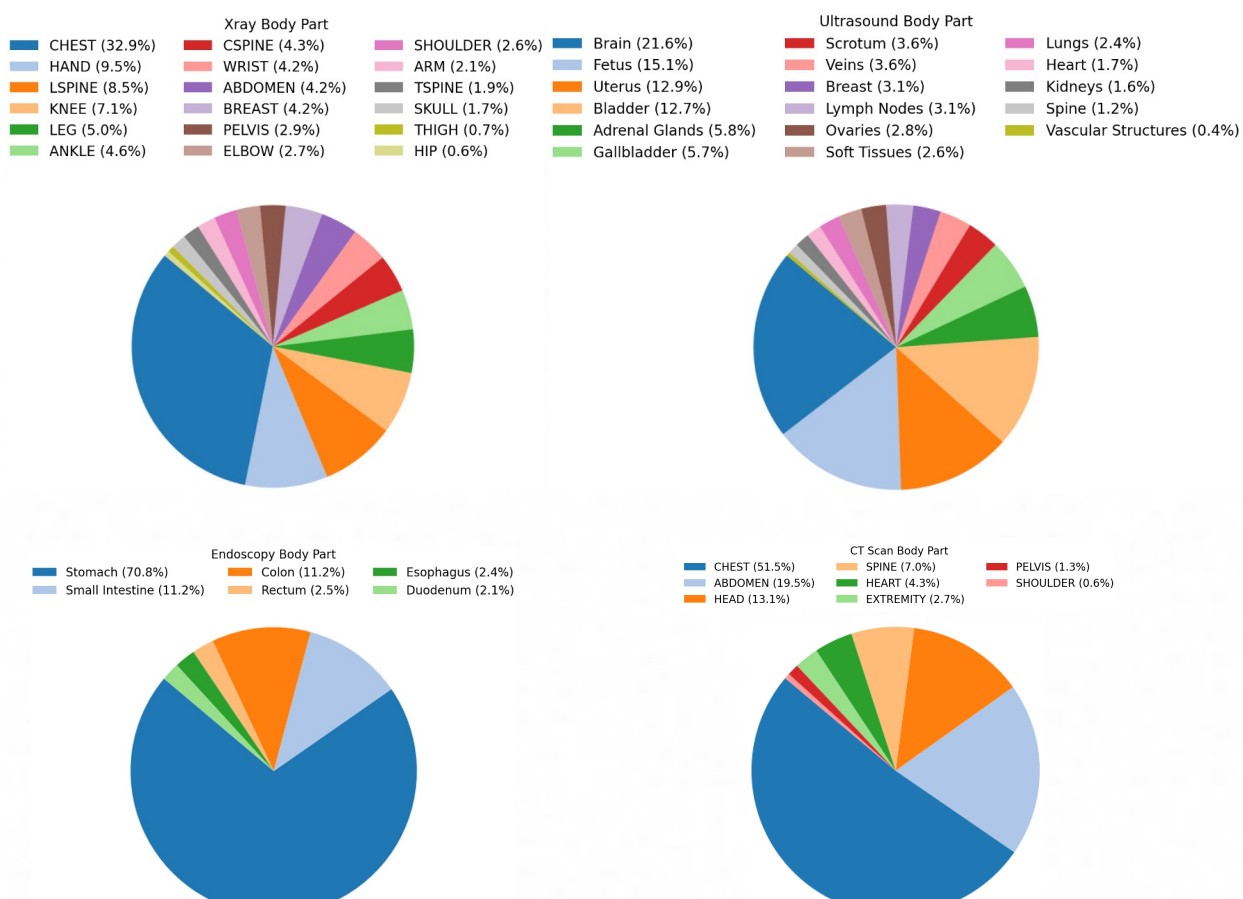

Figure 6: Distribution of body parts in the pretraining dataset. **Top Left:** X-ray. **Top Right:** Ultrasound. **Bottom Left:** Endoscopy. **Bottom Right:** CT.

Table 5: Sample distribution across four imaging modalities in our collected dataset.

| Total | X-ray | Ultrasound | CT | Endoscopy |
|---|---|---|---|---|
| 867,653 | 286,240 | 299,698 | 233,088 | 48,627 |

# B  Details of Data Augmentation in SimDINO

---

**Algorithm 1:** Data Augmentation for 2D Images

---

**Input:** Image $X \in \mathbb{R}^{1 \times H \times W}$, global crop scale $s_g$, local crop scale $s_l$, number of local crops $N_l$
**Output:** A list of augmented crops: $\{X^{(g1)}, X^{(g2)}, X^{(l1)}, \ldots, X^{(lN_l)}\}$
Define global crop size $S_g = (s_g \cdot H, s_g \cdot W)$
Define local crop size $S_l = (s_l \cdot H, s_l \cdot W)$
**Transformations:**
    Global Transform1: RandCrop $\rightarrow$ Resize $\rightarrow$ RandFlip $\rightarrow$ RandShiftIntensity $\rightarrow$ Gaussian blur $\rightarrow$ Normalize
    Global Transform2: RandCrop $\rightarrow$ Resize $\rightarrow$ RandFlip $\rightarrow$ RandShiftIntensity $\rightarrow$ Contrast adjust $\rightarrow$
      Normalize
    Local Transform: RandCrop $\rightarrow$ Resize $\rightarrow$ Normalize
**Apply augmentations:**
$X^{(g1)} \leftarrow$ `global_transform1`$(X)$
$X^{(g2)} \leftarrow$ `global_transform2`$(X)$
**for** $i = 1$ **to** $N_l$ **do**
    $X^{(li)} \leftarrow$ `local_transform`$(X)$
Expand channel to $C = 3$ after each crop
**return** $\{X^{(g1)}, X^{(g2)}, X^{(l1)}, \ldots, X^{(lN_l)}\}$

---

**Algorithm 2:** Data Augmentation for Videos

---

**Input:** Video $X \in \mathbb{R}^{3 \times H \times W \times S}$, global crop scale $s_g$, local crop scale $s_l$, number of local crops $N_l$
**Output:** Augmented crops list
Get number of frames: $S \leftarrow \text{shape}(X)[2]$
**if** $S > 16$ **then**
    Randomly sample 16 indices: `indices` $\leftarrow \text{randint}(0, S, 16)$
    Temporal slice: $X \leftarrow X[:, :, :, \text{indices}]$
Define spatial crop sizes: $S_g = (s_g \cdot H, s_g \cdot W, -1)$, $S_l = (s_l \cdot H, s_l \cdot W, -1)$
Apply the same transformation logic as in `Data Augmentation for 2D Images`, but preserve the temporal axis
  $(S)$
**Return** two global and $N_l$ local augmented views

---

---

**Algorithm 3:** Data Augmentation for 3D Volumes

---

**Input:** Volume $X \in \mathbb{R}^{1 \times H \times W \times S}$, crop scales $s_g, s_l$, number of local crops $N_l$
**Output:** A list of augmented crops
Set appropriate sizes:
    Global crop size: $S_g = (s_g \cdot H, s_g \cdot W, s_g \cdot S)$
    Local crop size: $S_l = (s_l \cdot H, s_l \cdot W, s_l \cdot S)$
    Resize targets vary per case
Define transformation pipelines:
    Global Transform1: RandCrop $\rightarrow$ Resize $\rightarrow$ RandFlip $\rightarrow$ RandShiftIntensity $\rightarrow$ Gaussian blur $\rightarrow$ Normalize
    Global Transform2: RandCrop $\rightarrow$ Resize $\rightarrow$ RandFlip $\rightarrow$ RandShiftIntensity $\rightarrow$ Contrast adjust $\rightarrow$
     Normalize
    Local Transform: RandCrop $\rightarrow$ Resize $\rightarrow$ Normalize
**Apply augmentations:**
$X^{(g1)} \leftarrow$ `global_transform1`$(X)$
$X^{(g2)} \leftarrow$ `global_transform2`$(X)$
**for** $i = 1$ **to** $N_l$ **do**
    $X^{(li)} \leftarrow$ `local_transform`$(X)$
Expand channel to $C = 3$ after each crop
**return** $\{X^{(g1)}, X^{(g2)}, X^{(l1)}, \dots, X^{(lN_l)}\}$

---

The data augmentation method for 2D images is designed for grayscale modalities such as X-ray and ultrasound. It generates two global views and multiple local crops. Each transformation includes spatial cropping, optional noise or contrast adjustments, and normalization. The augmented image is then duplicated across the channel dimension to form three channels and expanded along the slice dimension to match the shared input format.

The data augmentation method for RGB endoscopy videos, which is similar to the 2D case in spatial dimensions, processes a temporal stack of frames while preserving the slice dimension (typically $S = 16$). The augmentations are applied only to the spatial axes.

The data augmentation method for 3D volumes of varying depth. It chooses transformation parameters and resizing targets accordingly. All volumes are normalized and augmented in 3D. After transformation, the single-channel input is expanded to three channels to align with the shared encoder input convention.

## C  Downstream Task Configuration

### C.1  Dataset Setup

In this study, we utilize several publicly available and private datasets to evaluate the performance of our models across various medical imaging modalities. The detailed specifications for each dataset are shown in Table 6, including its composition, the number of images used for testing, and the organs or disease categories it encompasses.

**Category-level Retrieval:** For this task, we use the following public datasets: ChestXray14 Wang et al. (2017) for X-ray modality, Fetal Planes Burgos-Artizzu et al. (2020) for Ultrasound, and Kvasir Capsule Smedsrud et al. (2020) and Hyper Kvasir Borgli et al. (2020) for Endoscopy.

- ChestX-ray14 Wang et al. (2017) is a widely recognized medical imaging dataset that comprises 112,120 frontal-view X-ray images obtained from 30,805 unique patients. The dataset includes fourteen common thoracic disease labels, covering a diverse range of conditions, including Atelectasis, Effusion, Infiltration, Pneumothorax, Edema, Emphysema, Fibrosis, Pleural Thickening, Cardiomegaly, Nodule, Mass, Hernia, Consolidation, and Pneumonia. To ensure clarity and focus in our evaluation, we excluded images associated with multiple abnormal categories, resulting in a refined test set of 7,992 samples. This selection process aimed to minimize label ambiguity and facilitate a more precise assessment of the model's diagnostic capabilities.

- Fetal Planes Burgos-Artizzu et al. (2020) is a comprehensive collection of routinely acquired maternal-fetal screening ultrasound images, sourced from two distinct hospitals. Each image in the dataset has been meticulously annotated by expert maternal-fetal clinicians and categorized into six classes: Abdomen, Brain, Femur, Thorax, Cervix, and Other. For the purpose of our evaluation, we excluded the "Other" class to refine the dataset and constructed a test set comprising 8,188 images. This curated test set was used to assess the model's ability to accurately classify fetal anatomical planes, ensuring a focused and meaningful evaluation of its performance.

- Kvasir Capsule Smedsrud et al. (2020) stands as the largest publicly available PillCAM dataset in the field of gastrointestinal imaging. The dataset comprises 47,238 labeled images and spans a total of 117 videos, capturing anatomical landmarks, pathological findings, and normal observations. For our experiments, we processed the labeled images by grouping consecutive frames into video segments. Specifically, we aggregated sequential images to form video clips, ensuring that each clip contained at least 16 frames. Finally, we obtained a total of 344 video segments, which were subsequently used for our evaluations.

- HyperKvasir Borgli et al. (2020) is a comprehensive image and video dataset capturing the gastrointestinal tract. The dataset comprises 373 videos featuring various findings and anatomical landmarks. Each video was manually reviewed by medical professionals specializing in gastroenterology, resulting in a total of 171 annotated findings. For our study, we utilized Qwen2.5 Bai et al. (2025) to classify the findings into 13 distinct categories. We excluded any data that could not be reliably categorized, ultimately constructing a refined test set containing 364 samples.

**Progressive Regional Retrieval Tasks:** To evaluate the model's ability to handle fine-grained regional information, we design three increasingly challenging tasks, all using the RadGenome-ChestCT Zhang et al. (2024) dataset.

- RadGenome-ChestCT Zhang et al. (2024) is a large-scale, region-guided 3D chest CT interpretation dataset based on the CT-RATE Hamamci et al. (2024) dataset, which is the first publicly available dataset that pairs 3D medical images with corresponding textual reports. RadGenome-ChestCT extends the original CT-RATE of over 25,692 non-contrast 3D chest CT volumes and reports from more than 20,000 patients by incorporating organ-level segmentation masks covering 197 categories, multi-granularity grounded reports linked to specific anatomical regions, and 1.3 million grounded Visual Question Answering (VQA) pairs. These enhancements provide rich intermediate reasoning visual clues and enable models to associate textual explanations with corresponding visual evidence. In our experiments, we evaluated the model on a test set of 25,687 CT images to assess its performance in interpreting complex regional information. To ensure label quality, we employed Qwen2.5 Bai et al. (2025) as an auxiliary annotator to clean and harmonize the original labels, resolving inconsistencies and ambiguities in the annotations. This process yielded a refined taxonomy of 21 well-defined abnormality categories, which were subsequently used for performance evaluation.

**Cross-modal Retrieval:** In addition to the public datasets, we also utilized private datasets for cross-modal retrieval tasks. From the private MRI test set, we sampled 6,000 images representing 12 body parts. Similarly, we sampled 4,500 CT images encompassing 9 body parts and 9,500 X-ray images covering 19 body parts from the respective private test sets. These private datasets allowed us to evaluate the model's generalizability across diverse imaging modalities and anatomical structures.

**Downstream Classification:** To further evaluate the quality and transferability of the learned representations, we conduct downstream classification tasks on two datasets previously introduced: the publicly available RadGenome-ChestCT Zhang et al. (2024) for multi-label disease classification (18 categories) and a private MRI dataset for organ classification (12 categories). For the RadGenome-ChestCT dataset, we follow the data split protocol established in Zhang et al. (2024), which consists of 24,123 training samples and 1,564 test samples. This split ensures a consistent and fair comparison with prior work. For the private MRI dataset, we perform a stratified train-test split with a 4:1 ratio to maintain class distribution balance

Table 6: Overview of downstream datasets used in the study.

| Task Category | Dataset | Modality | Test Size | Classes/Regions |
|---|---|---|---|---|
| Category-level Retrieval | ChestXray14 | X-ray | 7,992 | 14 classes |
| | Fetal Planes | Ultrasound | 8,188 | 5 anatomical planes |
| | Kvasir Capsule | Endoscopy | 344 | 13 abnormalities |
| | Hyper Kvasir | Endoscopy | 364 | 13 abnormalities |
| Progressive Regional Retrieval | CT-RATE | CT | 25,687 | 21 abnormalities |
| Cross-modal Retrieval | Private MRI | MRI | 6,000 | 12 body parts |
| | Private CT | CT | 4,500 | 9 body parts |
| | Private X-ray | X-ray | 9,500 | 19 body parts |
| Downstream Classification | CT-RATE | CT | 1,564 | 18 abnormalities |
| | Private MRI | MRI | 1,200 | 12 body parts |

across both subsets, resulting in a dedicated training set and an independently held-out test set for unbiased evaluation. All models are trained on the respective training sets using frozen features to assess the effectiveness of the pre-trained representations in a linear probing setup.

By leveraging these datasets, we aim to provide a comprehensive evaluation of our models' capabilities in handling medical imaging data across various modalities and pathological contexts.

## C.2 Implementation Details

With the pretrained vision encoder, we compute the embeddings for each data sample, then we calculate the cosine similarity between the query and candidate samples to evaluate their relevance. In intra-modal retrieval, every test sample is used as a query and the gallery is the remaining samples from the same test set; the query item itself is always removed before ranking. In cross-modal retrieval, queries come from one modality and the gallery comes from the target modality. The recall list is then examined to determine whether the retrieved samples belong to the same disease category or anatomy, thereby verifying if they are true positive samples. Labels are used only after retrieval to score the ranked list, and are not used for embedding extraction, similarity computation, or ranking.

When a dataset provides patient, study, video, or image identifiers, we use those identifiers to de-duplicate the evaluation split and avoid placing the same underlying case in both query and gallery as distinct entries. We do not apply additional heuristic near-duplicate filtering beyond dataset-provided identifiers because such rules are difficult to define consistently across X-ray, CT, ultrasound, and endoscopy. Public evaluation datasets are not used during pretraining. For private evaluation datasets, including the cross-modal retrieval benchmark, we split and de-duplicate by patient or study identifiers before evaluation, and no patient or study identifier appears in both the pretraining set and the private test set.

Specifically, in the context of anatomy-level cross-modal retrieval, a key challenge arises from the fact that different modalities have different descriptions about varying field-of-views or anatomical body parts, typically reflected by the "BodyPartExamined" tag within the DICOM (Digital Imaging and Communications in Medicine) format. To correlate images depicting the same body part across different modalities, we leverage Qwen2.5 Yang et al. (2024) only at evaluation time to establish a tag mapping between anatomical regions across different modalities. This mapping is used to assign positives and negatives after retrieval, and Qwen2.5 is not used during model training, embedding extraction, similarity computation, or retrieval ranking.

For lesion-size retrieval, continuous lesion-size annotations are discretized into fixed size buckets before evaluation. A retrieved candidate is counted as correct only if it matches the query on the relevant anatomical region or abnormality and falls into the same lesion-size bucket. For endoscopy videos, 16 frames are randomly sampled during training as temporal augmentation. During evaluation, we uniformly sample 16 frames once over the full video duration and use this deterministic clip to compute the video embedding.

# D  Statistical Uncertainty and Additional Baselines

For each method and task, we compute 95% bootstrap confidence intervals using 1,000 query-level bootstrap replicates. In each replicate, evaluation queries are resampled with replacement, the retrieval gallery is fixed after self-match removal, and R@1, R@5, R@10, MnR, and MdR are recomputed. We report each result as point estimate $\pm$ 95% CI half-width, computed from the 2.5th and 97.5th percentiles of the bootstrap distribution. Results with overlapping intervals or very small margins should be interpreted as comparable rather than clear improvements.

Table 7: Category-level retrieval with 95% bootstrap CI on ChestXray14 and Fetal Planes.

| Model | ChestXray14 | | | | | Fetal Planes | | | | |
|---|---|---|---|---|---|---|---|---|---|---|
| | R@1 | R@5 | R@10 | MnR | MdR | R@1 | R@5 | R@10 | MnR | MdR |
| DINOv2-B | 0.311 ±0.011 | 0.658 ±0.008 | 0.798 ±0.008 | 6.277 ±0.068 | 6.246 ±0.067 | 0.890 ±0.007 | 0.982 ±0.003 | 0.992 ±0.002 | 5.462 ±0.017 | 5.444 ±0.019 |
| DINOv3-7B | 0.304 ±0.010 | 0.649 ±0.010 | 0.793 ±0.009 | 6.277 ±0.065 | 6.250 ±0.066 | 0.939 ±0.005 | 0.994 ±0.002 | 0.997 ±0.001 | 5.482 ±0.011 | 5.474 ±0.013 |
| SigLIP2[†] | 0.247 ±0.009 | 0.599 ±0.010 | 0.760 ±0.009 | 6.636 ±0.070 | 6.631 ±0.072 | 0.902 ±0.006 | 0.987 ±0.002 | 0.995 ±0.002 | 5.465 ±0.015 | 5.449 ±0.017 |
| BMC-CLIP[†] | 0.284 ±0.009 | 0.631 ±0.011 | 0.777 ±0.010 | 6.479 ±0.065 | 6.460 ±0.066 | 0.947 ±0.005 | 0.991 ±0.002 | 0.996 ±0.001 | 5.484 ±0.013 | 5.474 ±0.015 |
| UniMiSS+ | 0.298 ±0.010 | 0.648 ±0.010 | 0.783 ±0.008 | 6.384 ±0.068 | 6.354 ±0.072 | / | / | / | / | / |
| M³Ret (MAE) | 0.247 ±0.010 | 0.603 ±0.010 | 0.762 ±0.010 | 6.575 ±0.068 | 6.552 ±0.068 | 0.881 ±0.008 | 0.966 ±0.005 | 0.981 ±0.003 | 5.429 ±0.029 | 5.399 ±0.031 |
| M³Ret (SimDINO) | 0.345 ±0.010 | 0.674 ±0.010 | 0.812 ±0.008 | 6.136 ±0.068 | 6.095 ±0.069 | 0.955 ±0.006 | 0.990 ±0.002 | 0.996 ±0.001 | 5.456 ±0.014 | 5.445 ±0.016 |

Table 8: Category-level retrieval with 95% bootstrap CI on Kvasir Capsule and HyperKvasir.

| Model | Kvasir Capsule | | | | | HyperKvasir | | | | |
|---|---|---|---|---|---|---|---|---|---|---|
| | R@1 | R@5 | R@10 | MnR | MdR | R@1 | R@5 | R@10 | MnR | MdR |
| DINOv2-B | 0.784 ±0.042 | 0.898 ±0.032 | 0.936 ±0.026 | 5.132 ±0.225 | 5.038 ±0.243 | 0.396 ±0.048 | 0.668 ±0.049 | 0.736 ±0.043 | 6.788 ±0.286 | 6.801 ±0.287 |
| DINOv3-7B | 0.776 ±0.045 | 0.913 ±0.031 | 0.939 ±0.026 | 5.191 ±0.206 | 5.106 ±0.218 | 0.387 ±0.047 | 0.673 ±0.049 | 0.731 ±0.048 | 6.850 ±0.291 | 6.865 ±0.293 |
| SigLIP2[†] | 0.720 ±0.048 | 0.895 ±0.034 | 0.927 ±0.028 | 5.245 ±0.240 | 5.134 ±0.256 | 0.374 ±0.047 | 0.657 ±0.051 | 0.731 ±0.048 | 6.775 ±0.288 | 6.755 ±0.297 |
| BMC-CLIP[†] | 0.743 ±0.048 | 0.904 ±0.032 | 0.936 ±0.026 | 5.186 ±0.232 | 5.117 ±0.249 | 0.390 ±0.048 | 0.654 ±0.051 | 0.706 ±0.048 | 6.842 ±0.295 | 6.817 ±0.297 |
| M³Ret (MAE) | 0.606 ±0.050 | 0.843 ±0.039 | 0.901 ±0.034 | 5.360 ±0.269 | 5.289 ±0.281 | 0.412 ±0.048 | 0.687 ±0.049 | 0.753 ±0.047 | 6.592 ±0.286 | 6.563 ±0.293 |
| M³Ret (SimDINO) | 0.647 ±0.051 | 0.872 ±0.038 | 0.913 ±0.035 | 5.302 ±0.271 | 5.201 ±0.278 | 0.448 ±0.048 | 0.690 ±0.049 | 0.747 ±0.045 | 6.617 ±0.296 | 6.657 ±0.299 |

Table 9: CT regional abnormality and lesion-size retrieval with 95% bootstrap CI.

| Model | Regional Abnormality | | | | | Lesion Size | | | | |
|---|---|---|---|---|---|---|---|---|---|---|
| | R@1 | R@5 | R@10 | MnR | MdR | R@1 | R@5 | R@10 | MnR | MdR |
| CT-FM | 0.044 ±0.003 | 0.173 ±0.005 | 0.273 ±0.005 | 9.494 ±0.035 | 9.496 ±0.035 | 0.007 ±0.002 | 0.031 ±0.004 | 0.059 ±0.005 | 10.666 ±0.034 | 10.666 ±0.034 |
| CT-CLIP | 0.038 ±0.002 | 0.150 ±0.004 | 0.242 ±0.005 | 9.652 ±0.033 | 9.653 ±0.033 | 0.004 ±0.001 | 0.022 ±0.004 | 0.045 ±0.005 | 10.760 ±0.030 | 10.760 ±0.030 |
| VoCo | 0.036 ±0.002 | 0.150 ±0.004 | 0.241 ±0.005 | 9.663 ±0.033 | 9.663 ±0.033 | 0.004 ±0.001 | 0.021 ±0.004 | 0.043 ±0.005 | 10.773 ±0.031 | 10.773 ±0.031 |
| UniMiSS+ | 0.040 ±0.002 | 0.159 ±0.005 | 0.255 ±0.005 | 9.592 ±0.034 | 9.593 ±0.034 | 0.006 ±0.002 | 0.028 ±0.004 | 0.050 ±0.005 | 10.711 ±0.034 | 10.711 ±0.034 |
| Merlin | 0.054 ±0.003 | 0.198 ±0.005 | 0.303 ±0.006 | 9.311 ±0.035 | 9.313 ±0.035 | 0.010 ±0.002 | 0.043 ±0.005 | 0.076 ±0.007 | 10.571 ±0.039 | 10.572 ±0.039 |
| M³Ret (MAE) | 0.050 ±0.003 | 0.188 ±0.005 | 0.295 ±0.006 | 9.378 ±0.035 | 9.376 ±0.035 | 0.012 ±0.003 | 0.037 ±0.005 | 0.070 ±0.006 | 10.598 ±0.038 | 10.597 ±0.038 |
| M³Ret (SimDINO) | 0.058 ±0.003 | 0.200 ±0.005 | 0.303 ±0.005 | 9.298 ±0.035 | 9.299 ±0.035 | 0.014 ±0.003 | 0.046 ±0.005 | 0.076 ±0.006 | 10.544 ±0.043 | 10.544 ±0.043 |

Table 10: Anatomy-level cross-modal retrieval with 95% bootstrap CI.

| Model | CT → MRI | | | | | MRI → CT | | | | |
|---|---|---|---|---|---|---|---|---|---|---|
| | R@1 | R@5 | R@10 | MnR | MdR | R@1 | R@5 | R@10 | MnR | MdR |
| UniMiSS+ | 0.160 ±0.012 | 0.303 ±0.015 | 0.370 ±0.016 | 9.126 ±0.084 | 9.142 ±0.084 | 0.207 ±0.010 | 0.315 ±0.011 | 0.380 ±0.012 | 8.802 ±0.074 | 8.790 ±0.075 |
| VoCo | 0.258 ±0.014 | 0.561 ±0.017 | 0.676 ±0.016 | 7.115 ±0.101 | 7.056 ±0.106 | 0.196 ±0.010 | 0.342 ±0.011 | 0.424 ±0.012 | 8.692 ±0.071 | 8.677 ±0.074 |
| Merlin | 0.370 ±0.016 | 0.626 ±0.016 | 0.710 ±0.015 | 7.006 ±0.100 | 6.977 ±0.102 | 0.235 ±0.011 | 0.463 ±0.013 | 0.547 ±0.012 | 7.895 ±0.079 | 7.871 ±0.081 |
| $M^3$Ret (MAE) | 0.301 ±0.016 | 0.525 ±0.017 | 0.641 ±0.016 | 7.552 ±0.105 | 7.554 ±0.105 | 0.246 ±0.011 | 0.329 ±0.011 | 0.397 ±0.012 | 8.661 ±0.068 | 8.591 ±0.071 |
| $M^3$Ret (SimDINO) | 0.424 ±0.014 | 0.607 ±0.016 | 0.677 ±0.017 | 7.285 ±0.099 | 7.274 ±0.100 | 0.252 ±0.011 | 0.495 ±0.013 | 0.581 ±0.012 | 7.746 ±0.076 | 7.767 ±0.077 |

| Model | CT → X-ray | | | | | X-ray → CT | | | | |
|---|---|---|---|---|---|---|---|---|---|---|
| | R@1 | R@5 | R@10 | MnR | MdR | R@1 | R@5 | R@10 | MnR | MdR |
| UniMiSS+ | 0.204 ±0.012 | 0.336 ±0.013 | 0.417 ±0.014 | 8.778 ±0.082 | 8.774 ±0.081 | 0.344 ±0.010 | 0.518 ±0.010 | 0.595 ±0.010 | 7.745 ±0.059 | 7.720 ±0.061 |
| $M^3$Ret (MAE) | 0.327 ±0.010 | 0.513 ±0.010 | 0.572 ±0.011 | 7.837 ±0.057 | 7.799 ±0.057 | 0.310 ±0.007 | 0.552 ±0.008 | 0.625 ±0.009 | 7.403 ±0.055 | 7.396 ±0.055 |
| $M^3$Ret (SimDINO) | 0.262 ±0.012 | 0.487 ±0.014 | 0.548 ±0.014 | 7.912 ±0.079 | 7.900 ±0.082 | 0.285 ±0.009 | 0.502 ±0.010 | 0.581 ±0.010 | 7.851 ±0.057 | 7.836 ±0.057 |

| Model | MRI → X-ray | | | | | X-ray → MRI | | | | |
|---|---|---|---|---|---|---|---|---|---|---|
| | R@1 | R@5 | R@10 | MnR | MdR | R@1 | R@5 | R@10 | MnR | MdR |
| UniMiSS+ | 0.012 ±0.003 | 0.044 ±0.006 | 0.073 ±0.007 | 10.599 ±0.045 | 10.597 ±0.045 | 0.101 ±0.008 | 0.153 ±0.010 | 0.192 ±0.010 | 9.895 ±0.065 | 9.890 ±0.065 |
| $M^3$Ret (MAE) | 0.055 ±0.003 | 0.162 ±0.006 | 0.217 ±0.006 | 9.802 ±0.037 | 9.792 ±0.039 | 0.130 ±0.005 | 0.200 ±0.009 | 0.242 ±0.010 | 9.718 ±0.063 | 9.717 ±0.061 |
| $M^3$Ret (SimDINO) | 0.152 ±0.008 | 0.306 ±0.012 | 0.373 ±0.013 | 8.787 ±0.091 | 8.798 ±0.093 | 0.165 ±0.008 | 0.382 ±0.011 | 0.482 ±0.011 | 8.383 ±0.070 | 8.388 ±0.070 |

Table 11: Modality-specific specialist baseline comparison. These results illustrate specialization versus unification: specialist encoders can be strongest in their target modality, whereas $M^3$Ret uses one visual-only encoder and retrieval protocol across modalities.

| Model | Fetal Planes | | | | | Kvasir Capsule / HyperKvasir | | | | |
|---|---|---|---|---|---|---|---|---|---|---|
| | R@1 | R@5 | R@10 | MnR | MdR | Kvasir R@1 | Kvasir R@5 | Hyper R@1 | Hyper R@5 | Hyper R@10 |
| DINOv2-B | 0.890 | 0.982 | 0.992 | 5.462 | 5.444 | 0.784 | 0.898 | 0.396 | 0.668 | 0.736 |
| DINOv3-7B | 0.939 | 0.994 | 0.997 | 5.482 | 5.474 | 0.776 | 0.913 | 0.387 | 0.673 | 0.731 |
| SigLIP2 | 0.902 | 0.987 | 0.995 | 5.465 | 5.449 | 0.720 | 0.895 | 0.374 | 0.657 | 0.731 |
| BMC-CLIP | 0.947 | 0.991 | 0.996 | 5.484 | 5.474 | 0.743 | 0.904 | 0.390 | 0.654 | 0.706 |
| FetalCLIP | 0.963 | 0.998 | 0.998 | 5.527 | 5.529 | / | / | / | / | / |
| EndoViT | / | / | / | / | / | 0.901 | 0.985 | 0.234 | 0.492 | 0.563 |
| $M^3$Ret (MAE) | 0.881 | 0.966 | 0.981 | 5.429 | 5.399 | 0.606 | 0.843 | 0.412 | 0.687 | 0.753 |
| $M^3$Ret (SimDINO) | 0.955 | 0.990 | 0.996 | 5.456 | 5.445 | 0.647 | 0.872 | 0.448 | 0.690 | 0.747 |

Table 12: Role of large general-domain and biomedical baselines. M3Ret (SimDINO) is the proposed medical encoder trained with a DINO-style objective, not an external DINOv2 or DINOv3 checkpoint.

| Model | Model scale | Pretraining data scale | Supervision / data type |
|---|---|---|---|
| $M^3$Ret | ViT-B, ~86M parameters | 867,653 medical imaging samples | Visual-only SSL; X-ray, ultrasound, endoscopy video, CT |
| DINOv2-B | ViT-B, ~86M parameters | 142M natural images | Visual-only SSL; natural images |
| DINOv3-7B | ~6.7B parameters | ~1.7B natural images | Visual-only SSL; natural images |
| BMC-CLIP | ViT-L, ~307M parameters | 24,076,288 image-caption pairs | Language-supervised biomedical image-text pairs |

# E Downstream Classification Performance

To further validate the quality and transferability of the learned representations, we evaluate $M^3$Ret on downstream classification tasks as a supplementary transfer check. We benchmark our model on two distinct datasets: CT-RATE Hamamci et al. (2024) for multi-label disease classification (18 categories) and

Table 13: Comparison of classification performance on the CT-RATE and MRI datasets. The best and second-best results are highlighted in **bold** and underlined, respectively. † denotes models using language supervision, and ‡ and § indicate supervised pretraining with segmentation masks and disease categories.

| Method | CT-RATE | | | | In-house MRI | | | |
|---|---|---|---|---|---|---|---|---|
| | F1 Score ↑ | Precision ↑ | Accuracy ↑ | AUROC ↑ | F1 Score ↑ | Precision ↑ | Accuracy ↑ | AUROC ↑ |
| VoCo‡ Wu et al. (2024a) | 0.105 | 0.259 | 0.810 | 0.636 | 0.427 | 0.404 | 0.404 | 0.847 |
| UniMiSS+ Xie et al. (2024) | 0.156 | 0.472 | 0.810 | 0.702 | 0.828 | 0.828 | 0.829 | 0.982 |
| CT-FM Pai et al. (2025) | 0.306 | 0.494 | 0.803 | 0.763 | / | / | / | / |
| CT-CLIP† Hamamci et al. (2024) | 0.136 | 0.324 | 0.819 | 0.720 | / | / | / | / |
| Merlin§ Blanke-meier et al. (2024) | / | / | / | / | 0.948 | 0.948 | 0.949 | **0.997** |
| $M^3$Ret (MAE) | 0.178 | 0.355 | 0.824 | 0.749 | 0.785 | 0.782 | 0.785 | 0.964 |
| $M^3$Ret (SimDINO) | **0.313** | **0.624** | **0.840** | **0.797** | **0.954** | **0.954** | **0.955** | **0.997** |

our in-house MRI dataset for body part classification (12 body parts). As shown in Table 13, $M^3$Ret with SimDINO pretraining achieves strong results across F1 Score, Precision, Accuracy, and AUROC. This suggests that SimDINO learns more discriminative features than MAE in this setting. Interestingly, Merlin also performs well on the MRI task despite having never been trained on MRI data. This suggests that its cross-modal generalization capability may be largely attributed to learning robust representations of anatomical structures, which are transferable across modalities. Note that CT-FM Pai et al. (2025) and CT-CLIP Hamamci et al. (2024) were not evaluated on the MRI task, as their pretraining is limited to CT data. Similarly, Merlin was excluded from the CT-RATE evaluation, as its supervised pretraining with disease labels would create an unfair comparison with the other self-supervised methods. These results provide supplementary evidence beyond retrieval, rather than a comprehensive evaluation of downstream generality.

## F  Limitations and Future Work

We pretrain $M^3$Ret on mainstream medical imaging modalities, including 2D grayscale images (X-ray, ultrasound), RGB videos (endoscopy), and 3D volumetric scans (CT), which cover a broad range of clinical scenarios. However, the medical imaging domain encompasses a wider variety of modalities such as PET, SPECT, functional MRI (fMRI), and whole-slide histopathology images, each with distinct characteristics and clinical value. Even within CT imaging, different acquisition protocols, such as varying radiation doses or contrast phases, may introduce distribution shifts. In addition, CT volumes are resized or padded to a fixed 64-slice voxel grid without enforcing a common physical voxel spacing or isotropic resampling. As a result, the model does not explicitly encode absolute physical scale, which may affect lesion-size retrieval and motivates future spacing-aware resampling or physical-coordinate embeddings. Due to the high cost, limited availability, and data heterogeneity associated with these modalities, collecting large-scale, high-quality datasets for unified pretraining remains a major challenge. In future work, we aim to extend our framework to incorporate more diverse and fine-grained modalities, and explore transfer strategies to adapt unified representations to data-scarce or rare imaging types.

## G  More Visualization Results

We present additional results for regional abnormality retrieval and anatomy-level cross-modal retrieval tasks. Our proposed model, $M^3$Ret, achieves strong performance across most configurations.

The results of regional abnormality retrieval are illustrated in Figure 7. Our approach achieves these results using purely self-supervised learning (SSL), without relying on any task-specific design or annotations. This is a significant advancement, as it demonstrates the capability of generic visual SSL to implicitly capture localized pathological cues, which are often critical for fine-grained medical image analysis. By leveraging self-supervision, our method avoids the need for region-level supervision, which is typically labor-intensive and requires expert annotations. Besides this, the model demonstrates strong performance in retrieving fine-grained details, suggesting that the learned representations inherently encode meaningful regional information. This ability to capture localized features without explicit guidance highlights the potential of SSL to serve as a powerful tool for medical imaging tasks, where annotated data is often scarce or expensive to obtain.

The results of anatomy-level cross-modal retrieval are shown in Figures 8, 9, 10, 11, 12, 13. $M^3$Ret demonstrates strong generalization capabilities to MRI-related tasks, despite never having been exposed to MRI data during pretraining. This highlights the model's ability to transfer visual knowledge across diverse medical imaging modalities without requiring modality-specific fine-tuning or explicit domain adaptation. These findings suggest that self-supervised learning (SSL) in the visual domain can learn visual representations that support anatomy-level organization across specific imaging modalities. Furthermore, our unified SSL framework encodes diverse imaging modalities such as CT, X-ray, and MRI into a shared latent space without relying on paired samples or explicit cross-modal alignment strategies, which are often resource-intensive and challenging to obtain in real-world scenarios. These results provide evidence for leveraging self-supervised learning in multimodal medical retrieval applications, where labeled data is scarce or difficult to acquire.

In summary, our work contributes to advancing medical image retrieval by demonstrating that a single, unified SSL framework can effectively handle multiple imaging modalities under a common visual encoder and retrieval protocol. This may support retrieval-oriented research workflows in settings with heterogeneous datasets.

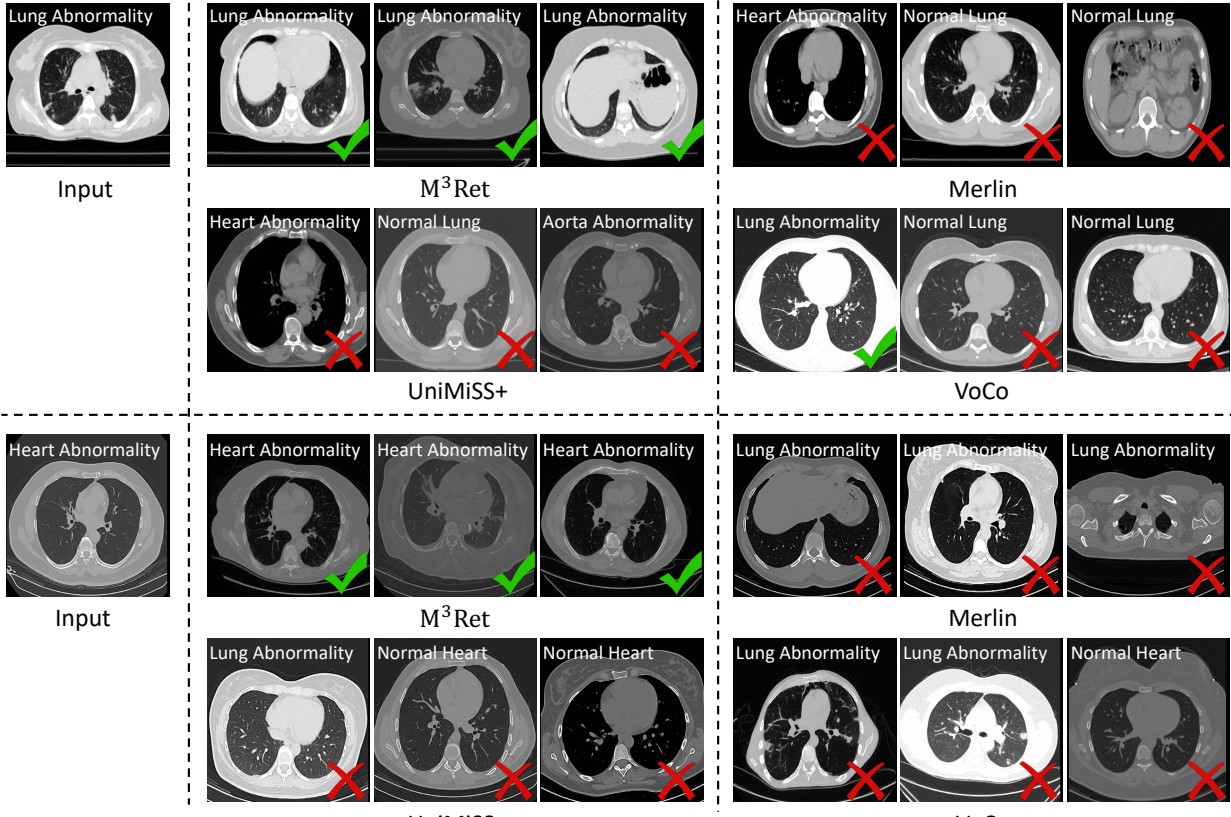

Figure 7: **Qualitative results of top 3 retrieval examples in regional abnormality retrieval.**

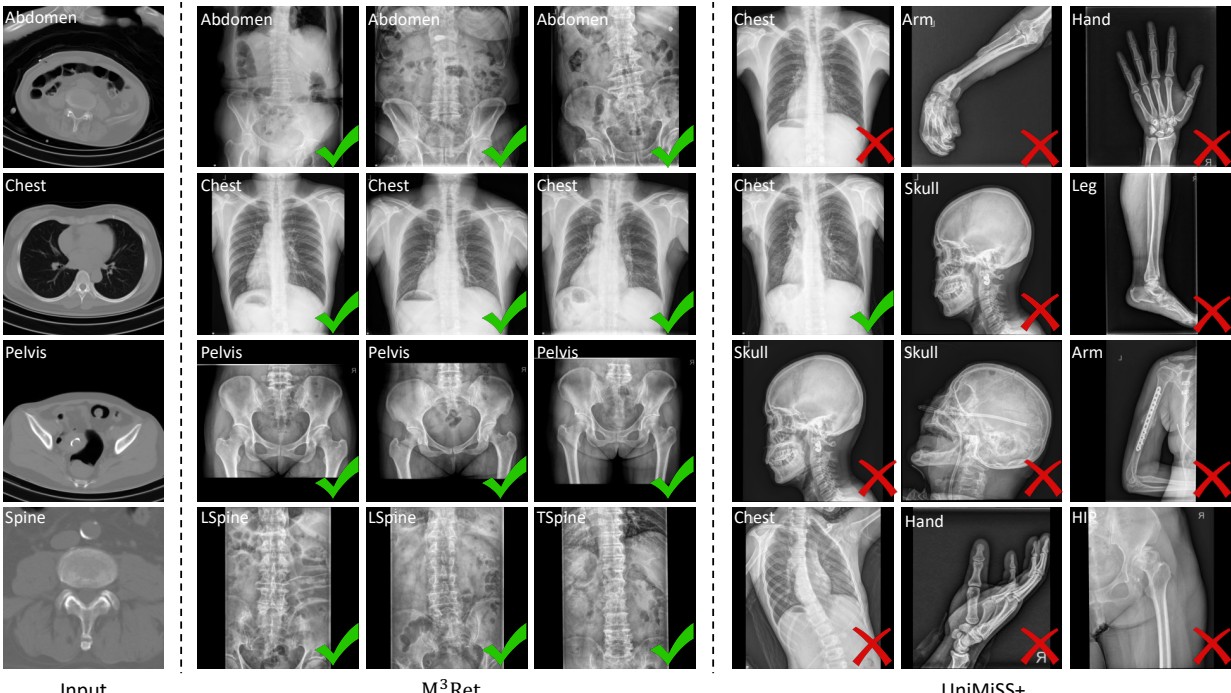

Figure 8: **Qualitative results of top 3 retrieval examples in CT → X-ray Task.**

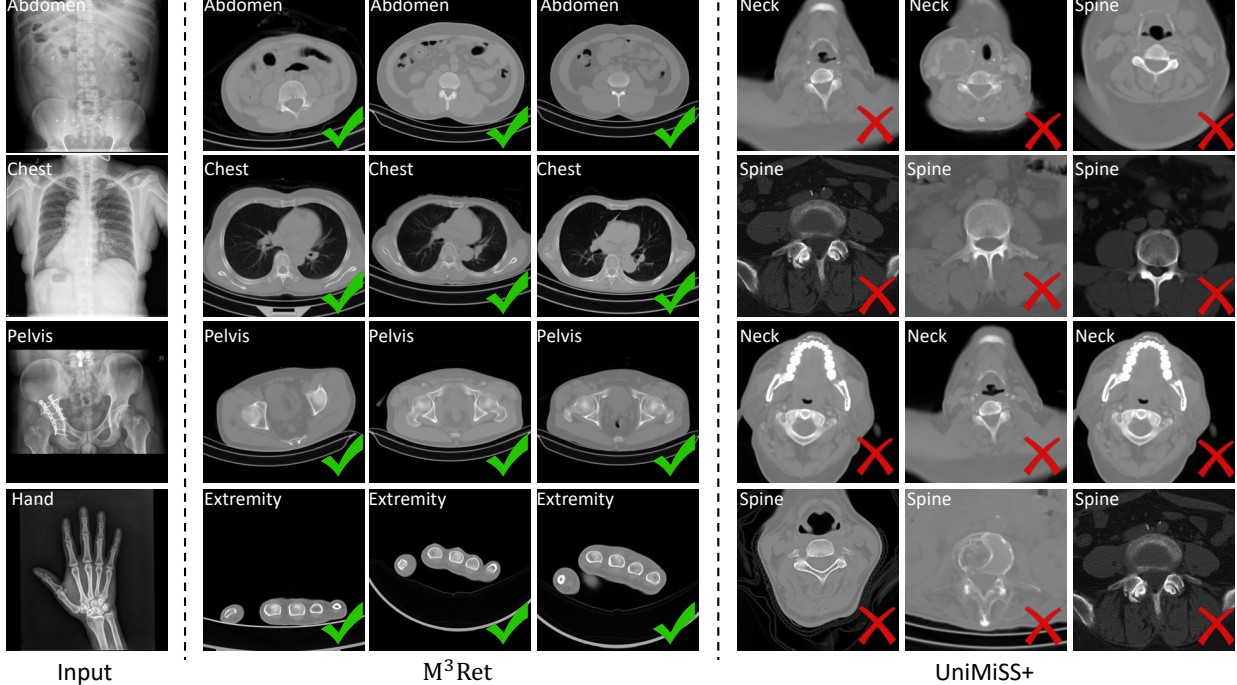

Figure 9: **Qualitative results of top 3 retrieval examples in X-ray → CT Task.**

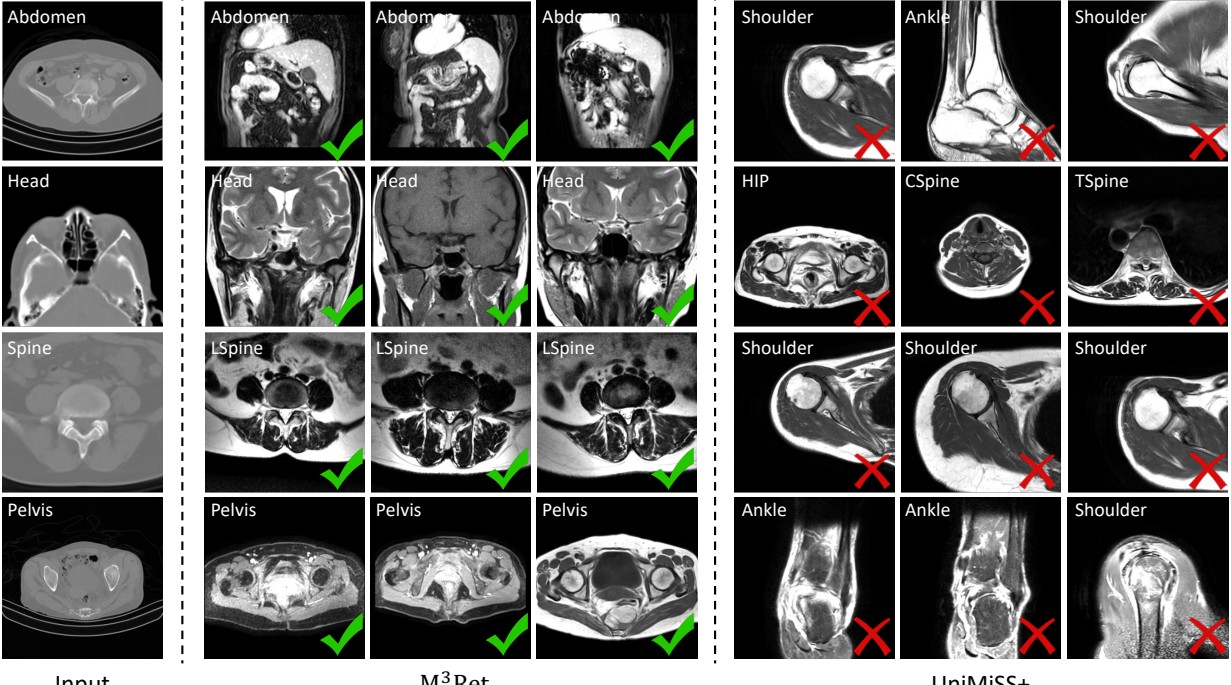

Figure 10: **Qualitative results of top 3 retrieval examples in CT → MRI Task.**

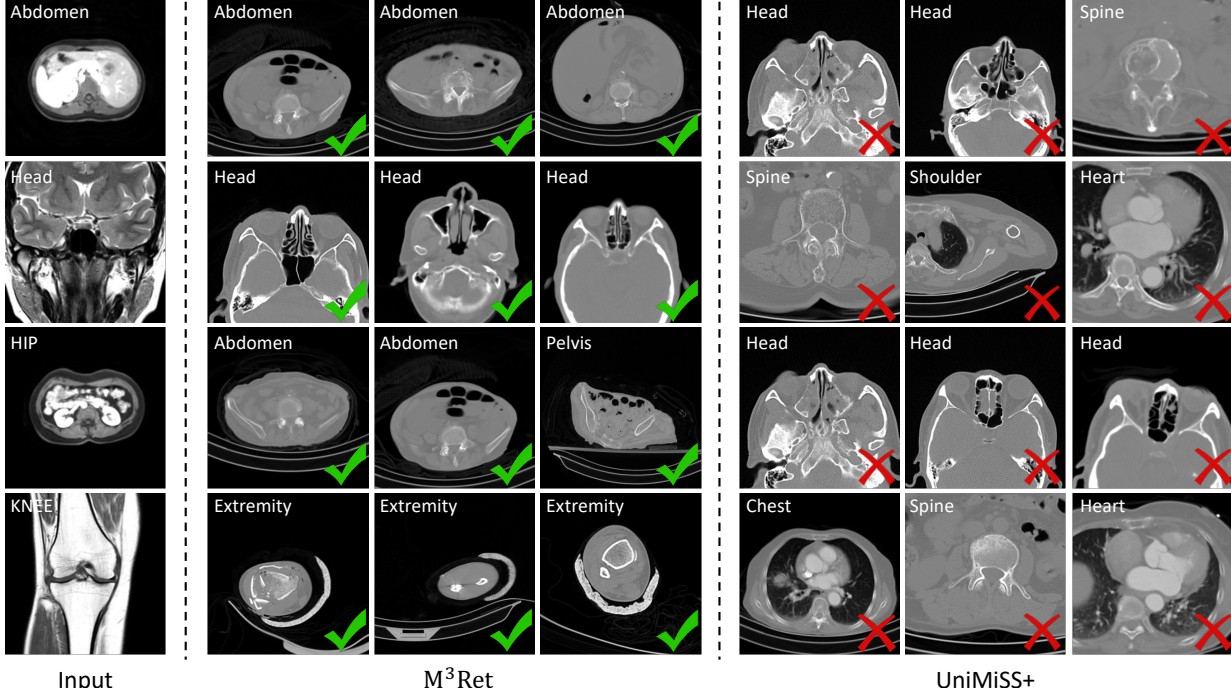

Figure 11: **Qualitative results of top 3 retrieval examples in MRI → CT Task.**

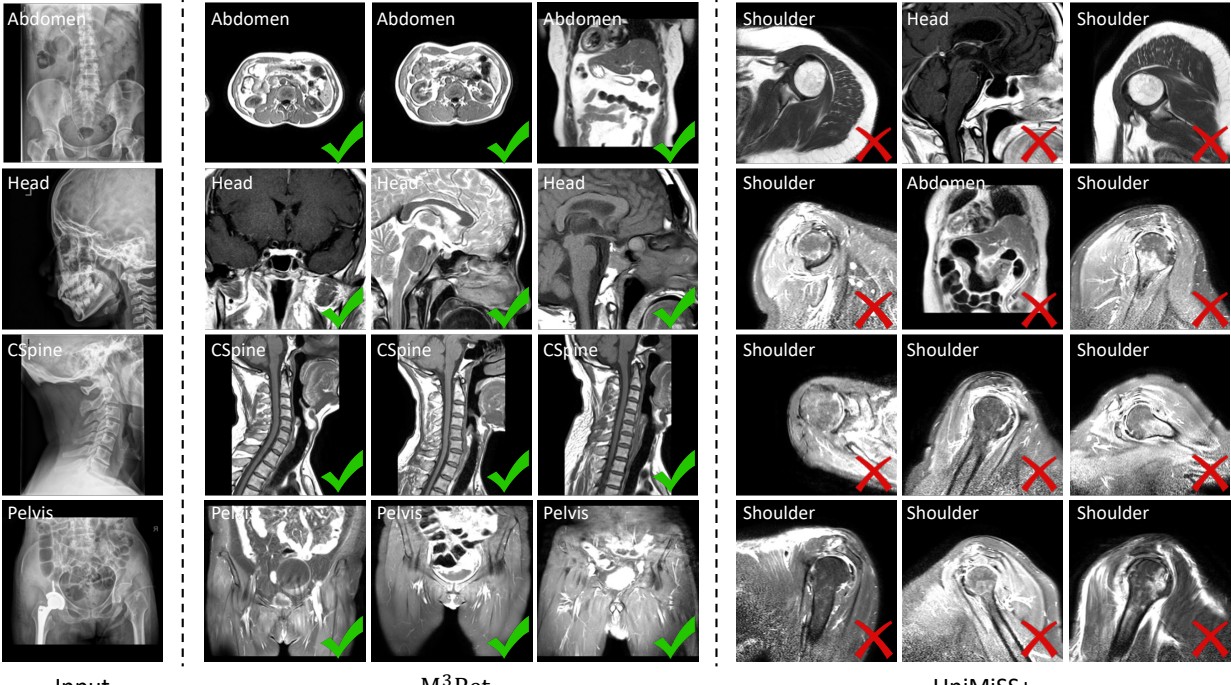

Figure 12: **Qualitative results of top 3 retrieval examples in X-ray → MRI Task.**

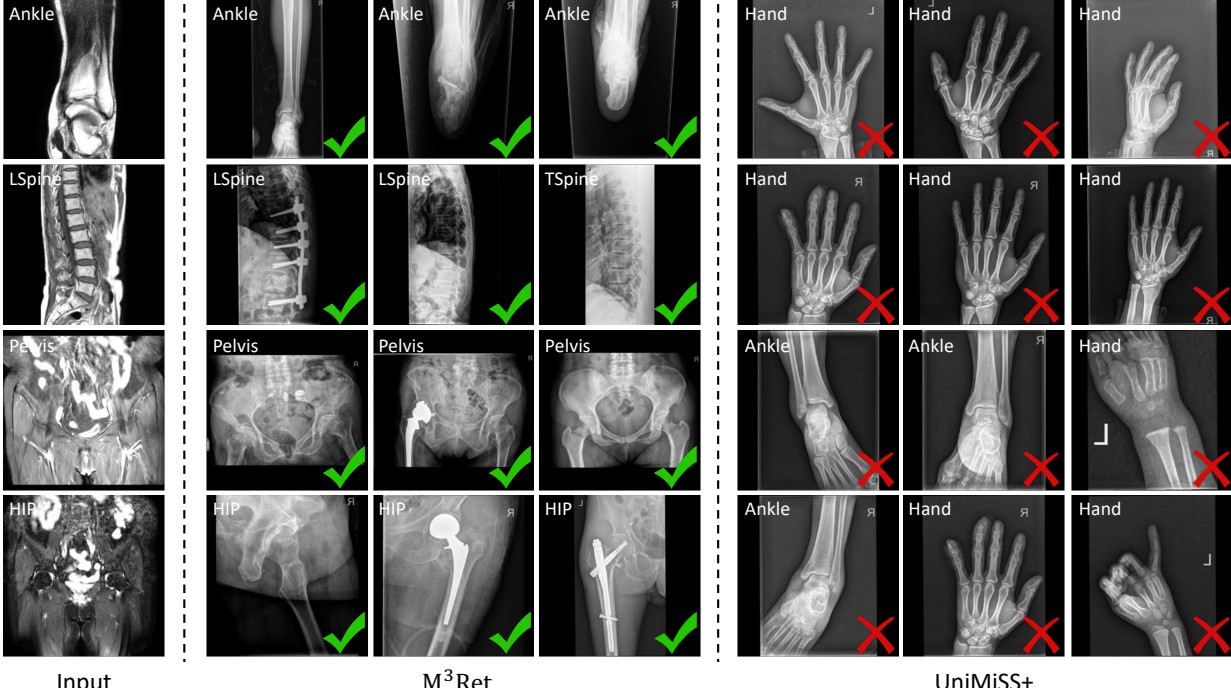

Figure 13: **Qualitative results of top 3 retrieval examples in MRI → X-ray Task.**

