# OpenReview forum: "M3Ret: Unleashing Zero-shot Multi-Modal Medical Image Retrieval via Self-Supervision"
_TMLR — Accepted by TMLR_

### Review · Reviewer_skmW · 2026-03-30

**Summary Of Contributions:**

This paper presents a method for medical image retrieval across different modalities using self-supervised learning (SSL). The authors introduce a new dataset containing various image types, including 2D X-rays and ultrasound images, 3D CT scans, and endoscopy videos, covering various anatomical regions. The authors then train ViTs on this dataset, with two different SSL methods: MAE and SimDINO. The models are trained on image data only, without text or segmentation labels. The learned representations are then used for both intra- and cross-modal retrieval tasks, showing higher accuracy than other methods.

**Audience:**

Yes

**Audience Explanation:**

Researchers on image retrieval, multi-modal learning, and foundation models for medical images would be interested.

**Broader Impact Concerns:**

None.

**Claims And Evidence:**

No

**Claims Explanation:**

The claim that this work shows "visual SSL enables scalable and effective representation learning across heterogeneous medical imaging modalities" is a bit too broad. The representation is only evaluated for retrieval in this paper, but not for other downstream tasks such as classification or segmentation. The authors may want to focus on the retrieval problem and avoid making broader claims on representation learning.

**Requested Changes:**

Strengths:
- The paper targets a challenging problem: retrieval across different medical imaging modalities. This is an important problem and aligns with the current trends toward multi-modal models.
- The results obtained are encouraging and could be informative to researchers on foundation models for medical images.

Weaknesses:
- The novelty of methodology is relatively limited: The paper mainly applies existing SSL methods to a more diverse dataset.
- Although the method is motivated by cross-modal retrieval, the evaluation is somehow focused on intra-modal retrieval, with limited results on cross-modal retrieval.
- Some details on evaluation metrics and method design can be better clarified.

Detailed comments:
- What's the correctness criterion for "cross-modal retrieval"? The authors write "Given a query image, the task is to retrieve an image from a different modality that shares the same category or semantic content." What does "category" and "semantic content" refer to specifically? Anatomical region, disease, or abnormality? Can the authors clarify what diseases are included in the dataset?

- In many experimental results, the retrieved image is deemed correct as long as it shares the same anatomical region as the query image. This seems not particularly relevant to clinical practice, since it is quite straightforward to find images of a specified body part--either from public datasets or through image metadata. Can the authors provide more insights on why an expensive deep learning model is necessary for this goal of retrieving images of the same anatomical region?

- "Each query image is labeled with regional lesion descriptions (e.g., “abdomen” with “hypodense lesion, 20 mm”). A retrieval is correct if the returned image matches the lesion size annotations for the relevant regions."
Can the authors clarify on what "match" means here? Must the retrieved image have a lesion of the exact same size (20 mm)? Or is there some tolerance?

- "Each query image is labeled with regional abnormality statuses (e.g., the “aorta” labeled as “normal” or “abnormal”)."
Does the query embedding only encode the image content, or the abnormality label as well? Can the authors clarify if and how the abnormality label is included in the query embedding?

- For endoscopy data, the authors "randomly sample S = 16 frames from a single video". Can the authors clarify why subsampling is necessary or helpful? Why did the authors use random sampling instead of sampling at a fixed interval? The randomly sampled frames may not be representative of the original video content.

- Can the authors explain the choice of 4 as the third dimension of the patch size? How is this value determined?

---

> ### Author Response · Authors · 2026-05-21
>
> Thank you for the constructive review. We will revise the manuscript to narrow the main claim, clarify the methodological contribution, make the cross-modal evaluation protocol explicit, and add missing details on lesion-size matching, label usage, video-frame sampling, and temporal/depth patchification.
>
> ### W1 / R1: Scope of the main claim
>
> We thank the reviewer for pointing out that the original claim may read as broader than the evidence directly supports. We will narrow the main evidence to zero-shot medical image retrieval across heterogeneous modalities, which serves as the primary evaluation protocol for the learned visual representation.
>
> Our motivation for using retrieval is to evaluate the pretrained representation without introducing additional task-specific training choices, such as fine-tuning recipes, optimizer settings, decoder architectures, or dense annotation protocols. This allows a more consistent comparison of frozen representations across heterogeneous modalities.
>
> At the same time, we will highlight the supplementary linear-probing evidence in the Appendix as an additional transfer check beyond retrieval. This experiment keeps the encoder frozen and evaluates classification on two modalities: CT-RATE for CT multi-label disease classification and an in-house MRI dataset for body-part classification. We will present this as supplementary evidence only, rather than using it to claim comprehensive downstream generality.
>
> ### W2: Positioning the methodological novelty
>
> We will position the contribution around the visual SSL scaling study and the unified representation framework for heterogeneous medical imaging. The key design is a modality-agnostic 4D patchification interface with a shared ViT encoder, which enables 2D images, 3D volumes, and 2D+T videos to be processed under the same tokenizer and representation pipeline.
>
> We will revise the paper to emphasize that the main focus is scale and unification: how visual SSL training behaves as model size and data size increase, how MAE and SimDINO compare across modalities under the same framework, and how a unified visual encoder organizes heterogeneous modalities, including anatomy-level results involving MRI despite MRI not being included during pretraining. Retrieval is used as the evaluation protocol for probing the learned representation across same-modality and cross-modality settings, rather than being the sole methodological contribution.
>
> ### W3 / Q1 / Q2: Cross-modal retrieval scope and anatomy-level correctness
>
> We will clarify that the cross-modal retrieval results in Table 3 evaluate anatomy-level alignment, not pathology-level or finding-level retrieval. The evaluated cross-modal settings include CT-to-MRI, MRI-to-CT, CT-to-X-ray, X-ray-to-CT, MRI-to-X-ray, and X-ray-to-MRI retrieval. A retrieved sample is counted as correct only when the query and retrieved candidate are mapped to the same anatomical region or body part across modalities. For example, a chest CT query is evaluated against candidates mapped to the same chest/thoracic region, rather than against candidates with the same disease label.
>
> The anatomical labels are used only after retrieval to assign positives and negatives for evaluation. They are not provided to the encoder, are not used to compute embeddings, and are not used in the similarity ranking. Qwen2.5 is used only to harmonize inconsistent anatomical labels or DICOM body-part descriptions across datasets for evaluation-time label mapping. It is not used during model training, embedding extraction, similarity computation, or retrieval ranking.
>
> We will therefore revise vague wording such as "same category" or "same semantic content" to "same mapped anatomical region/body part" for the cross-modal setting. We will also explicitly state that Table 3 should be interpreted as evidence of anatomy-level cross-modal organization in the frozen visual representation, not as evidence of disease-level, finding-level, or same-patient cross-modal retrieval. The stronger goal of disease-level or finding-level cross-modal retrieval remains a limitation and future direction.
>
> ### W4 / R2 / Q3: Definition of lesion-size matching
>
> Lesion-size retrieval is evaluated using fixed size buckets, not exact millimeter-level matching. Continuous lesion-size annotations are discretized into predefined intervals before evaluation, and the same bucket assignment is used for all compared methods. A retrieved candidate is counted as correct only if it matches the query on the relevant anatomical region or abnormality and falls into the same lesion-size bucket.
>
> We will clarify this in the paper to avoid the impression that a candidate must exactly match a lesion size such as 20 mm. The important evaluation point is that the intervals are fixed before scoring and shared across all methods.

---

> > ### Author Response · Authors · 2026-05-21
> >
> > ### Q4: Whether labels are used in the query embedding
> >
> > No labels are used to form the query embedding or database embeddings. Each embedding is computed only from the input image, volume, or video using the visual encoder. Disease labels, abnormality labels, lesion-size labels, and body-part labels are used only after retrieval to determine whether each returned candidate is a true positive for the reported metric.
> >
> > ### Q5: Why sample 16 frames for endoscopy videos
> >
> > We use 16 frames to keep a fixed token budget across videos of different lengths and to make video batching computationally feasible. During training, frames are randomly sampled as temporal augmentation so that the model sees different temporal views of the same video. During evaluation, however, we uniformly sample 16 frames once over the full video duration and use this deterministic clip to compute the video embedding. This makes the reported retrieval metrics reproducible and avoids dependence on random evaluation-time frame draws.
> >
> > The choice of 16 frames also follows common video-transformer practice, where 16-frame clips are widely used to balance temporal coverage and computation, including in VideoMAE-style training protocols.
> >
> > ### Q6: Rationale for using temporal/depth patch size 4
> >
> > The temporal/depth patch size of 4 is a practical unified-tokenizer choice rather than a separately tuned modality-specific hyperparameter. In our 4D patchification interface, the temporal/depth dimension has canonical sizes of 4 for replicated 2D images, 16 for endoscopy videos, and 64 for CT volumes. A temporal/depth patch size of 4 exactly divides all three cases, producing one group for 2D images, four temporal groups for videos, and sixteen depth groups for CT volumes.
> >
> > This choice lets us use the same patchifier and shared ViT encoder across 2D images, 2D+T videos, and 3D volumes without modality-specific tokenizers or padding rules. A smaller temporal/depth patch size would increase sequence length and computation for videos and CT volumes, while a larger value would make video tokenization coarser and would not naturally handle the 2D-image setting. We will clarify that this value was chosen as a simple common setting that preserves temporal/depth structure while keeping training feasible.

---

### Review · Reviewer_XrYF · 2026-04-19

**Summary Of Contributions:**

This paper proposes a unified visual self-supervised pretraining framework for heterogeneous medical imaging modalities, 2D X-ray and ultrasound, endoscopy video, and 3D CT, using a single ViT encoder with a 4D spatiotemporal patchification. The pre-training leverages more than 860,000 samples from a collaborating hospital and applies two off-the-shelf SSL objectives, MAE and SimDINO. The evaluation spans category-level retrieval on X-ray/ultrasound/endoscopy, progressive regional-abnormality and lesion-size retrieval on RadGenome-ChestCT, cross-modal retrieval among CT, X-ray, and MRI with MRI unseen during pretraining, plus scaling/ablation studies.

### strengths

- The unified 4D patchification is a clean, genuinely modality-agnostic mechanism at the encoder level and a useful framing for medical SSL.
- The retrieval evaluation spans multiple regimes (category, regional-abnormality, lesion-size, cross-modal) and five modalities, which is broader than typical single-benchmark reporting.
- MRI zero-shot results are non-trivial.
- Multimodal pretraining does not degrade unimodal performance

### weakness
- The SOTA-superiority claim is contradicted by the paper's own Table 1 — DINOv3-7B beats M3Ret (SimDINO) on Kvasir Capsule at R@1/R@5/R@10 and BMC-CLIP / DINOv3-7B edge past M3Ret(SimDINO) on Fetal Planes R@5
- No confidence intervals, seeds, or significance tests are reported for Tables 1–3; this is blocking for Table 2 whose margins are within plausible query-level variance
- The "without any modality-specific design/customization" claim is broader than what is supported: slice depth differs by modality, local-view counts differ, batch sizes are modality-dependent, and sampling is modality-alternating.
- "Cross-modal semantic alignment" is evaluated only via Qwen2.5-mediated mapping, i.e., anatomy-level rather than finding- or pathology-level alignment

**Audience:**

Yes

**Audience Explanation:**

The unified 4D patchification recipe, the MRI zero-shot generalization result, the multimodal-pretraining-does-not-degrade-unimodal-performance finding, and the comparative behavior of MAE versus SimDINO across 2D/3D/video modalities are of direct interest to the TMLR readership working on medical-imaging SSL, foundation-model scaling, and multimodal representation learning.

**Broader Impact Concerns:**

I think the broader impact statement could be treated more seriously for a for a medical retrieval system rather than just claiming the underlying technologies "have not been associated with negative societal outcomes", introducing "no foreseeable risks," and asserting adherence to "strict data anonymization and privacy standards" without specifying the procedure.

**Claims And Evidence:**

No

**Claims Explanation:**

Please see the weaknesses.

**Requested Changes:**

1. Either provide additional evidence closing the Kvasir Capsule gap and the Fetal Planes R@5 gap, or narrow the claim.

2. Add variance estimates for the small-margin Table 2 results

3. Discuss the MAE > SimDINO reversal on CT$\leftrightarrow$X-ray in Table 3

4. Add at least one modality-specific SSL baseline per modality

---

> ### Author Response · Authors · 2026-05-21
>
> Thank you for the constructive review. We will revise the manuscript to address the over-broad performance claim, add uncertainty estimates, clarify the unified-tokenizer and cross-modal evaluation protocols, expand modality-specific baselines, and strengthen the broader-impact discussion.
>
> ### W1 / R1: SOTA-superiority claim and Kvasir Capsule / Fetal Planes gaps
>
> We thank the reviewer for pointing out that our original wording overstated the result. We will revise the manuscript to avoid claiming that M3Ret is state-of-the-art on every dataset and metric. The more precise conclusion is that M3Ret provides a single ViT-B-scale, visual-only SSL encoder that achieves strong zero-shot retrieval across several heterogeneous medical modalities, while some specialized or larger baselines remain stronger on particular datasets and metrics.
>
> In particular, we will explicitly discuss the exceptions in Table 1, including Kvasir Capsule and Fetal Planes R@5. We will replace phrases such as "sets a new state-of-the-art across all individual modalities" with "achieves strong and competitive zero-shot medical image retrieval across heterogeneous modalities." We will also clarify that comparisons to DINOv3-7B, DINOv2-B, and BMC-CLIP are intended to contextualize model scale, supervision, and pretraining source, not to imply uniform superiority.
>
> | Current expression | Issue raised by the reviewer | Refined expression |
> |---|---|---|
> | "sets a new state-of-the-art in zero-shot image-to-image retrieval across all individual modalities" | This can be read as claiming the best result on every dataset and every metric, while Table 1 has exceptions. | "achieves strong and competitive zero-shot medical image retrieval across heterogeneous modalities" |
> | "surpassing strong baselines such as DINOv3 and the text-supervised BMC-CLIP" | Too absolute because DINOv3 and BMC-CLIP are better on some Kvasir Capsule / Fetal Planes metrics. | "outperforming or matching strong baselines in several settings, while specialized or larger baselines remain stronger in some cases" |
> | "we outperform strong baselines such as DINOv3-7B" | Too broad unless tied to specific datasets and metrics. | "M3Ret outperforms DINOv3-7B on ChestXray14 and HyperKvasir, while specialized models remain strongest in some modality-specific settings" |
>
> We will also add model and data scale information to make the comparison more transparent:
>
> | Model | Model scale | Pretraining data scale | Supervision / data type | Role in comparison |
> |---|---:|---:|---|---|
> | M3Ret | ViT-B, approximately 86M parameters | 867,653 medical imaging samples | visual-only SSL; X-ray, ultrasound, endoscopy video, CT | Main method |
> | DINOv2-B | ViT-B, approximately 86M parameters | 142M natural images | visual-only SSL; natural images | Parameter-scale-matched general SSL baseline |
> | DINOv3-7B | approximately 6.7B parameters | approximately 1.7B natural images | visual-only SSL; natural images | Large-scale general visual SSL baseline |
> | BMC-CLIP | ViT-L, approximately 307M parameters | 24,076,288 image-caption pairs | language-supervised biomedical image-text pairs | Strong language-supervised biomedical baseline |
>
> This comparison is separate from M3Ret (SimDINO), which denotes our medical encoder trained with a DINO-style SSL objective, not an external DINOv2 or DINOv3 checkpoint.

---

> > ### Author Response · Authors · 2026-05-21
> >
> > ### W2 / R2: Missing uncertainty estimates for Tables 1-3
> >
> > We agree that uncertainty estimates are needed, especially because some margins in Tables 1-3 are small. We have computed 95% bootstrap confidence intervals for all retrieval metrics in Tables 1-3 and will include them in the revision.
> >
> > For each method and task, we use 1,000 query-level bootstrap replicates. In each replicate, we resample evaluation queries with replacement, keep the retrieval gallery fixed after self-match removal, and recompute R@1, R@5, R@10, MnR, and MdR. The confidence interval is computed from the 2.5th and 97.5th percentiles of the bootstrap distribution. Each bootstrap replicate samples the same number of queries as the original test set, regardless of whether the dataset has fewer or more than 1,000 queries.
> >
> > For compact table formatting, each result will be reported as point estimate ± 95% CI half-width. We will describe results with overlapping intervals or very small margins as comparable rather than as clear improvements. The revised table caption will also state that labels are used only for post-retrieval scoring and not for embedding extraction or similarity computation.
> >
> > Revised Tables 1-3 will be reported as follows:
> >
> > Table 1. Category-level retrieval with 95% bootstrap CI.
> >
> > ChestXray14 and Fetal Planes:
> >
> > | Model | Chest R@1 | Chest R@5 | Chest R@10 | Chest MnR | Chest MdR | Fetal R@1 | Fetal R@5 | Fetal R@10 | Fetal MnR | Fetal MdR |
> > |---|---:|---:|---:|---:|---:|---:|---:|---:|---:|---:|
> > | DINOv2-B | 0.311 ± 0.011 | 0.658 ± 0.010 | 0.798 ± 0.008 | 6.277 ± 0.068 | 6.246 ± 0.067 | 0.890 ± 0.007 | 0.982 ± 0.003 | 0.992 ± 0.002 | 5.462 ± 0.017 | 5.444 ± 0.019 |
> > | DINOv3-7B | 0.304 ± 0.010 | 0.649 ± 0.010 | 0.793 ± 0.009 | 6.277 ± 0.065 | 6.250 ± 0.066 | 0.939 ± 0.005 | 0.994 ± 0.002 | 0.997 ± 0.001 | 5.482 ± 0.011 | 5.474 ± 0.013 |
> > | SigLIP2† | 0.247 ± 0.009 | 0.599 ± 0.010 | 0.760 ± 0.009 | 6.636 ± 0.070 | 6.631 ± 0.072 | 0.902 ± 0.006 | 0.987 ± 0.002 | 0.995 ± 0.002 | 5.465 ± 0.015 | 5.449 ± 0.017 |
> > | BMC-CLIP† | 0.284 ± 0.009 | 0.631 ± 0.011 | 0.777 ± 0.010 | 6.479 ± 0.065 | 6.460 ± 0.066 | 0.947 ± 0.005 | 0.991 ± 0.002 | 0.996 ± 0.001 | 5.484 ± 0.013 | 5.474 ± 0.015 |
> > | UniMiSS+ | 0.298 ± 0.010 | 0.648 ± 0.010 | 0.783 ± 0.008 | 6.384 ± 0.068 | 6.354 ± 0.072 | / | / | / | / | / |
> > | M3Ret (MAE) | 0.247 ± 0.010 | 0.603 ± 0.011 | 0.762 ± 0.010 | 6.575 ± 0.068 | 6.552 ± 0.068 | 0.881 ± 0.008 | 0.966 ± 0.005 | 0.981 ± 0.003 | 5.429 ± 0.029 | 5.399 ± 0.031 |
> > | M3Ret (SimDINO) | 0.345 ± 0.010 | 0.674 ± 0.010 | 0.812 ± 0.008 | 6.136 ± 0.068 | 6.095 ± 0.069 | 0.955 ± 0.006 | 0.990 ± 0.002 | 0.996 ± 0.001 | 5.456 ± 0.014 | 5.445 ± 0.016 |
> >
> > Kvasir Capsule and HyperKvasir:
> >
> > | Model | Kvasir R@1 | Kvasir R@5 | Kvasir R@10 | Kvasir MnR | Kvasir MdR | HyperKvasir R@1 | HyperKvasir R@5 | HyperKvasir R@10 | HyperKvasir MnR | HyperKvasir MdR |
> > |---|---:|---:|---:|---:|---:|---:|---:|---:|---:|---:|
> > | DINOv2-B | 0.784 ± 0.042 | 0.898 ± 0.032 | 0.936 ± 0.026 | 5.132 ± 0.225 | 5.038 ± 0.243 | 0.396 ± 0.048 | 0.668 ± 0.049 | 0.736 ± 0.043 | 6.788 ± 0.286 | 6.801 ± 0.287 |
> > | DINOv3-7B | 0.776 ± 0.045 | 0.913 ± 0.031 | 0.939 ± 0.026 | 5.191 ± 0.206 | 5.106 ± 0.218 | 0.387 ± 0.047 | 0.673 ± 0.049 | 0.731 ± 0.048 | 6.850 ± 0.291 | 6.865 ± 0.293 |
> > | SigLIP2† | 0.720 ± 0.048 | 0.895 ± 0.034 | 0.927 ± 0.028 | 5.245 ± 0.240 | 5.134 ± 0.256 | 0.374 ± 0.047 | 0.657 ± 0.051 | 0.731 ± 0.048 | 6.775 ± 0.288 | 6.755 ± 0.297 |
> > | BMC-CLIP† | 0.743 ± 0.048 | 0.904 ± 0.032 | 0.936 ± 0.026 | 5.186 ± 0.232 | 5.117 ± 0.249 | 0.390 ± 0.048 | 0.654 ± 0.051 | 0.706 ± 0.048 | 6.842 ± 0.295 | 6.817 ± 0.297 |
> > | M3Ret (MAE) | 0.606 ± 0.050 | 0.843 ± 0.039 | 0.901 ± 0.034 | 5.360 ± 0.269 | 5.289 ± 0.281 | 0.412 ± 0.048 | 0.687 ± 0.049 | 0.753 ± 0.047 | 6.592 ± 0.286 | 6.563 ± 0.293 |
> > | M3Ret (SimDINO) | 0.647 ± 0.051 | 0.872 ± 0.038 | 0.913 ± 0.035 | 5.302 ± 0.271 | 5.201 ± 0.278 | 0.448 ± 0.048 | 0.690 ± 0.049 | 0.747 ± 0.045 | 6.617 ± 0.296 | 6.657 ± 0.299 |
> >
> > † denotes image-text pretrained baselines.

---

> > > ### Author Response · Authors · 2026-05-21
> > >
> > > Table 2. CT regional abnormality and lesion-size retrieval with 95% bootstrap CI.
> > >
> > > | Model | Regional R@1 | Regional R@5 | Regional R@10 | Regional MnR | Regional MdR | Lesion R@1 | Lesion R@5 | Lesion R@10 | Lesion MnR | Lesion MdR |
> > > |---|---:|---:|---:|---:|---:|---:|---:|---:|---:|---:|
> > > | CT-FM | 0.044 ± 0.003 | 0.173 ± 0.005 | 0.273 ± 0.005 | 9.494 ± 0.035 | 9.496 ± 0.035 | 0.007 ± 0.002 | 0.031 ± 0.004 | 0.059 ± 0.005 | 10.666 ± 0.034 | 10.666 ± 0.034 |
> > > | CT-CLIP | 0.038 ± 0.002 | 0.150 ± 0.004 | 0.242 ± 0.005 | 9.652 ± 0.033 | 9.653 ± 0.033 | 0.004 ± 0.001 | 0.022 ± 0.004 | 0.045 ± 0.005 | 10.760 ± 0.030 | 10.760 ± 0.030 |
> > > | VoCo | 0.036 ± 0.002 | 0.150 ± 0.004 | 0.241 ± 0.005 | 9.663 ± 0.033 | 9.663 ± 0.033 | 0.004 ± 0.001 | 0.021 ± 0.004 | 0.043 ± 0.005 | 10.773 ± 0.031 | 10.773 ± 0.031 |
> > > | UniMiSS+ | 0.040 ± 0.002 | 0.159 ± 0.005 | 0.255 ± 0.005 | 9.592 ± 0.034 | 9.593 ± 0.034 | 0.006 ± 0.002 | 0.028 ± 0.004 | 0.050 ± 0.005 | 10.711 ± 0.034 | 10.711 ± 0.034 |
> > > | Merlin | 0.054 ± 0.003 | 0.198 ± 0.005 | 0.303 ± 0.006 | 9.311 ± 0.035 | 9.313 ± 0.035 | 0.010 ± 0.002 | 0.043 ± 0.005 | 0.076 ± 0.007 | 10.571 ± 0.039 | 10.572 ± 0.039 |
> > > | M3Ret (MAE) | 0.050 ± 0.003 | 0.188 ± 0.005 | 0.295 ± 0.006 | 9.378 ± 0.035 | 9.376 ± 0.035 | 0.012 ± 0.003 | 0.037 ± 0.005 | 0.070 ± 0.006 | 10.598 ± 0.038 | 10.597 ± 0.038 |
> > > | M3Ret (SimDINO) | 0.058 ± 0.003 | 0.200 ± 0.005 | 0.303 ± 0.005 | 9.298 ± 0.035 | 9.299 ± 0.035 | 0.014 ± 0.003 | 0.046 ± 0.005 | 0.076 ± 0.006 | 10.544 ± 0.043 | 10.544 ± 0.043 |
> > >
> > > Table 3. Anatomy-level cross-modal retrieval with 95% bootstrap CI.
> > >
> > > CT to MRI and MRI to CT:
> > >
> > > | Model | CT-MRI R@1 | CT-MRI R@5 | CT-MRI R@10 | CT-MRI MnR | CT-MRI MdR | MRI-CT R@1 | MRI-CT R@5 | MRI-CT R@10 | MRI-CT MnR | MRI-CT MdR |
> > > |---|---:|---:|---:|---:|---:|---:|---:|---:|---:|---:|
> > > | UniMiSS+ | 0.160 ± 0.012 | 0.303 ± 0.015 | 0.370 ± 0.016 | 9.126 ± 0.084 | 9.142 ± 0.084 | 0.207 ± 0.010 | 0.315 ± 0.011 | 0.380 ± 0.012 | 8.802 ± 0.074 | 8.790 ± 0.075 |
> > > | VoCo | 0.258 ± 0.014 | 0.561 ± 0.017 | 0.676 ± 0.016 | 7.115 ± 0.101 | 7.056 ± 0.106 | 0.196 ± 0.010 | 0.342 ± 0.011 | 0.424 ± 0.012 | 8.692 ± 0.071 | 8.677 ± 0.074 |
> > > | Merlin | 0.370 ± 0.016 | 0.626 ± 0.016 | 0.710 ± 0.015 | 7.006 ± 0.100 | 6.977 ± 0.102 | 0.235 ± 0.011 | 0.463 ± 0.013 | 0.547 ± 0.012 | 7.895 ± 0.079 | 7.871 ± 0.081 |
> > > | M3Ret (MAE) | 0.301 ± 0.016 | 0.525 ± 0.017 | 0.641 ± 0.016 | 7.552 ± 0.105 | 7.554 ± 0.105 | 0.246 ± 0.011 | 0.329 ± 0.011 | 0.397 ± 0.012 | 8.661 ± 0.068 | 8.591 ± 0.071 |
> > > | M3Ret (SimDINO) | 0.424 ± 0.014 | 0.607 ± 0.016 | 0.677 ± 0.017 | 7.285 ± 0.099 | 7.274 ± 0.100 | 0.252 ± 0.011 | 0.495 ± 0.013 | 0.581 ± 0.012 | 7.746 ± 0.076 | 7.767 ± 0.077 |
> > >
> > > CT to X-ray and X-ray to CT:
> > >
> > > | Model | CT-X-ray R@1 | CT-X-ray R@5 | CT-X-ray R@10 | CT-X-ray MnR | CT-X-ray MdR | X-ray-CT R@1 | X-ray-CT R@5 | X-ray-CT R@10 | X-ray-CT MnR | X-ray-CT MdR |
> > > |---|---:|---:|---:|---:|---:|---:|---:|---:|---:|---:|
> > > | UniMiSS+ | 0.204 ± 0.012 | 0.336 ± 0.013 | 0.417 ± 0.014 | 8.778 ± 0.082 | 8.774 ± 0.081 | 0.344 ± 0.010 | 0.518 ± 0.010 | 0.595 ± 0.010 | 7.745 ± 0.059 | 7.720 ± 0.061 |
> > > | M3Ret (MAE) | 0.327 ± 0.010 | 0.513 ± 0.010 | 0.572 ± 0.011 | 7.837 ± 0.057 | 7.799 ± 0.057 | 0.310 ± 0.007 | 0.552 ± 0.008 | 0.625 ± 0.009 | 7.403 ± 0.055 | 7.396 ± 0.055 |
> > > | M3Ret (SimDINO) | 0.262 ± 0.012 | 0.487 ± 0.014 | 0.548 ± 0.014 | 7.912 ± 0.079 | 7.900 ± 0.082 | 0.285 ± 0.009 | 0.502 ± 0.010 | 0.581 ± 0.010 | 7.851 ± 0.057 | 7.836 ± 0.057 |
> > >
> > > MRI to X-ray and X-ray to MRI:
> > >
> > > | Model | MRI-X-ray R@1 | MRI-X-ray R@5 | MRI-X-ray R@10 | MRI-X-ray MnR | MRI-X-ray MdR | X-ray-MRI R@1 | X-ray-MRI R@5 | X-ray-MRI R@10 | X-ray-MRI MnR | X-ray-MRI MdR |
> > > |---|---:|---:|---:|---:|---:|---:|---:|---:|---:|---:|
> > > | UniMiSS+ | 0.012 ± 0.003 | 0.044 ± 0.006 | 0.073 ± 0.007 | 10.599 ± 0.045 | 10.597 ± 0.045 | 0.101 ± 0.008 | 0.153 ± 0.010 | 0.192 ± 0.010 | 9.895 ± 0.065 | 9.890 ± 0.065 |
> > > | M3Ret (MAE) | 0.055 ± 0.003 | 0.162 ± 0.006 | 0.217 ± 0.006 | 9.802 ± 0.037 | 9.792 ± 0.039 | 0.130 ± 0.005 | 0.200 ± 0.009 | 0.242 ± 0.010 | 9.718 ± 0.063 | 9.717 ± 0.061 |
> > > | M3Ret (SimDINO) | 0.152 ± 0.008 | 0.306 ± 0.012 | 0.373 ± 0.013 | 8.787 ± 0.091 | 8.798 ± 0.093 | 0.165 ± 0.008 | 0.382 ± 0.011 | 0.482 ± 0.011 | 8.383 ± 0.070 | 8.388 ± 0.070 |

---

> > > > ### Author Response · Authors · 2026-05-21
> > > >
> > > > ### W3: Clarifying the unified tokenizer and modality-specific settings
> > > >
> > > > We will clarify that "unified" refers to the tokenizer interface, encoder architecture, embedding extraction, and retrieval protocol, not to identical raw input dimensions for every modality. M3Ret does not use modality-specific encoders, modality-specific retrieval heads, modality-specific losses, or separate ranking modules. The differences in sampled depth, frame count, local-view count, and batch size are input standardization or computational scheduling choices needed to process 2D images, 3D volumes, and videos under a common token interface.
> > > >
> > > > Specifically, 2D images, 3D CT volumes, and endoscopy videos are converted to a 4D tensor interface and tokenized with the same patchifier shape, followed by the same ViT encoder and cosine-retrieval protocol. We will therefore replace "without any modality-specific design/customization" with "without modality-specific encoders, objectives, or retrieval heads." We will also include a clearer figure of the tokenizer pipeline in the revised manuscript.
> > > >
> > > > ### W4: Anatomy-level cross-modal retrieval vs pathology-level retrieval
> > > >
> > > > We agree that the cross-modal retrieval claim should be stated more precisely. Table 3 evaluates anatomy-level cross-modal retrieval, not pathology-level cross-modal diagnosis. Qwen2.5 is used only at evaluation time to harmonize heterogeneous anatomical labels across datasets for positive/negative assignment. It is not used during model training, embedding extraction, similarity computation, or retrieval ranking.
> > > >
> > > > For Table 3, a retrieved sample is counted as correct when the query and retrieved candidate correspond to the same mapped anatomical region/body part across modalities. Pathology-level retrieval is evaluated only where pathology annotations are available, such as disease-label retrieval for X-ray and regional abnormality or lesion-size retrieval for CT. We will revise "cross-modal semantic alignment" to "anatomy-level cross-modal retrieval" and will add finding-level cross-modal retrieval as an important limitation and future direction.
> > > >
> > > > ### R3: MAE greater than SimDINO on CT-X-ray in Table 3
> > > >
> > > > We will add a discussion of the CT-X-ray exception where MAE outperforms SimDINO. Although SimDINO performs better in most settings, CT-X-ray retrieval may require matching volumetric anatomical structure to 2D projection appearance. CT and radiography are both X-ray-based modalities, but they encode anatomy differently: CT reconstructs volumetric structure from multiple projections, whereas radiography compresses 3D anatomy into a 2D projection.
> > > >
> > > > We hypothesize that MAE's reconstruction objective may preserve global spatial and projection-relevant structure that is useful for this setting, whereas SimDINO's invariance-based objective may be more beneficial for other retrieval regimes. We will present this as a plausible hypothesis rather than a causal conclusion.
> > > >
> > > > ### R4: Modality-specific SSL baselines
> > > >
> > > > We agree that modality-specific medical SSL baselines are important. In the revision, we will expand the baseline discussion to include available modality-specific models where applicable, including CT-specific baselines and relevant X-ray, ultrasound, and endoscopy baselines. We will present these results as a specialization-versus-unification comparison: specialized encoders can be strongest in their target modality, while M3Ret aims to provide one visual-only encoder and retrieval protocol across X-ray, ultrasound, endoscopy, CT, and anatomy-level cross-modal retrieval.
> > > >
> > > > Ultrasound results:
> > > >
> > > > | Model | R@1 | R@5 | R@10 | MnR | MdR |
> > > > |---|---:|---:|---:|---:|---:|
> > > > | DINOv2-B | 0.923 | 0.983 | 0.993 | 5.435 | 5.414 |
> > > > | DINOv3-7B | 0.971 | 0.994 | 0.997 | 5.460 | 5.448 |
> > > > | SigLIP2 | 0.934 | 0.987 | 0.995 | 5.442 | 5.423 |
> > > > | BMC-CLIP | 0.947 | 0.991 | 0.996 | 5.460 | 5.447 |
> > > > | FetalCLIP | 0.963 | 0.998 | 0.998 | 5.527 | 5.529 |
> > > > | M3Ret (MAE) | 0.881 | 0.966 | 0.981 | 5.429 | 5.399 |
> > > > | M3Ret (SimDINO) | 0.955 | 0.990 | 0.996 | 5.456 | 5.445 |
> > > >
> > > > Endoscopy results:
> > > >
> > > > | Model | Kvasir R@1 | Kvasir R@5 | Kvasir R@10 | Kvasir MnR | Kvasir MdR | HyperKvasir R@1 | HyperKvasir R@5 | HyperKvasir R@10 | HyperKvasir MnR | HyperKvasir MdR |
> > > > |---|---:|---:|---:|---:|---:|---:|---:|---:|---:|---:|
> > > > | DINOv2-B | 0.784 | 0.898 | 0.936 | 5.132 | 5.038 | 0.396 | 0.668 | 0.736 | 6.788 | 6.801 |
> > > > | DINOv3-7B | 0.776 | 0.913 | 0.939 | 5.191 | 5.106 | 0.387 | 0.673 | 0.731 | 6.850 | 6.865 |
> > > > | SigLIP2 | 0.720 | 0.895 | 0.927 | 5.245 | 5.134 | 0.374 | 0.657 | 0.731 | 6.775 | 6.755 |
> > > > | BMC-CLIP | 0.743 | 0.904 | 0.936 | 5.186 | 5.117 | 0.390 | 0.654 | 0.706 | 6.842 | 6.817 |
> > > > | EndoViT | 0.901 | 0.985 | 0.994 | 4.532 | 4.343 | 0.234 | 0.492 | 0.563 | 7.491 | 7.474 |
> > > > | M3Ret (MAE) | 0.606 | 0.843 | 0.901 | 5.360 | 5.289 | 0.412 | 0.687 | 0.753 | 6.592 | 6.563 |
> > > > | M3Ret (SimDINO) | 0.647 | 0.872 | 0.913 | 5.302 | 5.201 | 0.448 | 0.690 | 0.747 | 6.617 | 6.657 |

---

> > > > > ### Author Response · Authors · 2026-05-21
> > > > >
> > > > > ### B1: Broader impact concern
> > > > >
> > > > > We agree and will revise the broader impact statement. We will remove "no foreseeable risks." The revised statement will note that retrieval outputs can be over-interpreted in clinical settings, that positive/negative pair definitions may be ambiguous when cases share labels but differ in subtype, severity, location, or partially overlapping findings, and that the system should not be used as an autonomous diagnostic tool.
> > > > >
> > > > > These risks motivate external and prospective validation, human-in-the-loop use, uncertainty reporting, and clear separation between research retrieval support and clinical decision-making. All private data were de-identified before model development and evaluation, and patient identifiers were not used as model inputs, retrieval keys, or similarity features.

---

### Review · Reviewer_SrqZ · 2026-05-09

**Summary Of Contributions:**

The paper proposes M3Ret, a unified visual SSL framework for medical image retrieval across X-ray, ultrasound, endoscopy video, and CT. The main idea is to represent different medical inputs with a common 4D patchification scheme and train a shared ViT encoder using MAE or SimDINO. The model is evaluated on intra-modal retrieval, CT regional retrieval, cross-modal retrieval, scaling analysis, and a small set of downstream classification tasks.

The paper addresses an important problem and the empirical scope is fairly broad. I find the unified encoder setup and the MRI zero-shot results interesting. However, some experimental details are still underspecified, especially the retrieval protocol and the effective pretraining schedule across modalities.

**Additional Comments:**

One minor point: the baseline choice for DINO is unclear. The paper reports DINOv2-B but DINOv3-7B. Please explain whether this was due to checkpoint availability or intended as a strongest-baseline comparison.

**Audience:**

Yes

**Audience Explanation:**

The paper is relevant to researchers working on medical image retrieval, self-supervised learning, and medical foundation models. Even if some claims need to be narrowed, the attempt to train a single visual SSL model over 2D images, 3D volumes, and videos is useful and timely.

**Claims And Evidence:**

No

**Claims Explanation:**

The results are promising, but I do not think the current evidence fully supports the broader claims.

My main concern is that the retrieval setup is not described carefully enough. For medical retrieval, patient-level, study-level, scan-level, and video-level overlap can strongly affect performance. The paper should be more explicit about how query and database images are separated, whether near-duplicates are removed, and whether clips from the same video or images from the same patient can appear on both sides of retrieval.

I am also not fully convinced by the training description. The modalities have different dataset sizes, token counts, and batch sizes, and the paper alternates modalities during training. Without the actual sampling probabilities or effective number of updates per modality, it is hard to know what is being compared or how to reproduce the setup.

**Requested Changes:**

- Clarify the retrieval protocol, including query/database construction, self-match removal, patient/study/scan/video-level separation, and near-duplicate handling.
- Provide more details on the mixed-modality pretraining schedule: modality sampling, effective epochs or updates per modality, and whether smaller modalities are oversampled.
- Clarify whether the private pretraining data overlaps with any private or public evaluation data at the patient, study, or institution level.
- Tone down the scaling-law discussion unless more runs or error bars are added. With only a few model/data sizes, this looks more like a preliminary scaling trend.
- Clarify CT preprocessing, especially whether voxel spacing is preserved or discarded when resizing volumes to a fixed number of slices.

---

> ### Author Response · Authors · 2026-05-21
>
> Thank you for the constructive review. We will revise the manuscript to make the retrieval protocol, mixed-modality pretraining schedule, pretraining/evaluation separation, scaling-language, CT preprocessing, and DINO baseline roles more explicit.
>
> ### W1 / R1: Retrieval protocol and possible overlap in query/database construction
>
> For retrieval evaluation, each test image or volume is embedded once by the frozen encoder, and nearest neighbors are ranked by embedding similarity. In intra-modal retrieval, every test sample is used as a query, and the gallery is the remaining samples from the same test set; the query item itself is always removed before ranking. In cross-modal retrieval, queries come from one modality and the gallery comes from the target modality.
>
> Class labels, abnormality labels, or body-part annotations are not used to form the query embedding or to rank candidates. They are used only after retrieval to determine whether each retrieved item is relevant for computing Recall@K and rank-based metrics.
>
> We prevent trivial matches in two ways. First, self-matches are removed from all intra-modal galleries. Second, when a dataset provides patient, study, video, or image identifiers, we use those identifiers to de-duplicate the evaluation split and avoid placing the same underlying case in both query and gallery as distinct entries. We did not apply heuristic near-duplicate filtering beyond dataset-provided identifiers because such filtering is difficult to define consistently across X-ray, CT, ultrasound, and endoscopy; we will state this explicitly as a limitation.
>
> Evaluation dataset summary:
>
> | Task Category | Dataset | Modality | Test Size | Classes/Regions |
> |---|---|---|---:|---|
> | Category-level retrieval | ChestXray14 | X-ray | 7,992 | 14 classes |
> | Category-level retrieval | Fetal Planes | Ultrasound | 8,188 | 5 anatomical planes |
> | Category-level retrieval | Kvasir Capsule | Endoscopy | 344 | 13 abnormalities |
> | Category-level retrieval | HyperKvasir | Endoscopy | 364 | 13 abnormalities |
> | Progressive regional retrieval | CT-RATE | CT | 25,687 | 21 abnormalities |
> | Cross-modal retrieval | Private MRI | MRI | 6,000 | 12 body parts |
> | Cross-modal retrieval | Private CT | CT | 4,500 | 9 body parts |
> | Cross-modal retrieval | Private X-ray | X-ray | 9,500 | 19 body parts |
>
> ### W2 / R2: Mixed-modality pretraining schedule
>
> During pretraining, we train on the union of all modalities for 300 epochs, where one epoch corresponds to one pass over the pooled pretraining set. We do not use modality-specific oversampling or target modality probabilities. For tensor collation, each GPU worker receives a local batch from a single modality with a consistent shape. Across workers at the same optimization step, different modalities may appear simultaneously, so the global batch can contain mixed modalities.
>
> The modality sample counts are 286,240 X-ray, 299,698 ultrasound, 233,088 CT, and 48,627 endoscopy samples. Local batch sizes are 32 for X-ray, ultrasound, and endoscopy, and 16 for CT. No modality is oversampled.
>
> | Modality | Samples | Local batch size | Sampling rule | Training duration | Oversampled? |
> |---|---:|---:|---|---:|---|
> | X-ray | 286,240 | 32 | same-modality local batches; no modality-specific oversampling | 300 epochs | No |
> | Ultrasound | 299,698 | 32 | same-modality local batches; no modality-specific oversampling | 300 epochs | No |
> | CT | 233,088 | 16 | same-modality local batches; no modality-specific oversampling | 300 epochs | No |
> | Endoscopy | 48,627 | 32 | same-modality local batches; no modality-specific oversampling | 300 epochs | No |
>
> ### W3 / R3: Overlap between pretraining and evaluation data
>
> There is no overlap between the pretraining set and the evaluation sets. Public evaluation datasets are not used during pretraining. For private evaluation datasets, including the cross-modal retrieval benchmark, we split and de-duplicate by patient/study identifiers before evaluation, and no patient or study identifier appears in both the pretraining set and the private test set.
>
> ### W4 / R4: Scaling-law wording
>
> We will revise the scaling-language to avoid overstating the evidence. Figure 4 provides preliminary empirical scaling trends, not a universal scaling law. Across the evaluated retrieval tasks, Recall@5 generally improves as the encoder increases from ViT-T to ViT-B and as the pretraining data ratio increases from 20% to 100%. We will describe these results as evidence that M3Ret benefits from increased model and data scale, rather than as proof of power-law scaling.
>
> | Current wording | Safer wording |
> |---|---|
> | "power-law scaling" / "scaling law" | "preliminary scaling trend" |
> | "validate the scalability" | "suggest that performance improves with scale" |
> | "clear answer" | "initial evidence" |

---

> > ### Author Response · Authors · 2026-05-21
> >
> > ### W5 / R5: CT preprocessing and voxel spacing
> >
> > For CT, we resize or pad each volume to a fixed voxel-grid shape with 64 slices, but we do not enforce a common physical voxel spacing or isotropic resampling. This keeps the token budget fixed across heterogeneous CT acquisitions, but it also means that absolute physical scale is not explicitly encoded.
> >
> > We agree this can matter for tasks such as lesion-size retrieval, and we will add this limitation together with future directions such as spacing-aware resampling or physical-coordinate embeddings.
> >
> > ### Q1: DINO baseline choice
> >
> > We will clarify the roles of the DINO baselines. DINOv2-B is included as a closer parameter-scale visual SSL baseline, while DINOv3-7B is included as a much larger general-purpose visual SSL reference. We do not present them as equivalent or scale-matched baselines. Their purpose is to compare M3Ret both against a roughly comparable encoder scale and against a substantially larger external foundation model.
> >
> > This is separate from SimDINO, which is the DINO-style SSL objective used to train our unified medical encoder rather than an external pretrained baseline.

---

### Decision · Action_Editor_unSu · 2026-06-29

**Recommendation:** Accept with minor revision

**Audience:**

Yes

**Audience Explanation:**

1. Unified image processing has broader computer vision relevance.

Although the paper focuses on medical imaging, the main question—whether a common 4D patchification scheme and a shared ViT encoder can process 2D images, 3D volumes, and videos—is also relevant to broader CV and self-supervised learning research. A unified visual SSL model that works across heterogeneous inputs is likely to be of interest to the broader CV community.

2. Comparisons with general-domain SSL models are valuable.

The comparisons with DINO-style and other general-domain SSL models help readers understand the strengths, limitations and failure modes of current visual SSL methods in vertical domains. The different behavior of MAE and SimDINO across 2D, 3D, and video data is also an interesting empirical observation for researchers working on visual representation learning.

3. The problem is relevant to multiple TMLR communities.

Retrieval across different medical imaging modalities is an important problem that is closely related to current research on multimodal foundation models. This work is likely to be of interest to researchers in image retrieval, multimodal learning, medical self-supervised learning, and medical foundation models.

**Claims And Evidence:**

Yes

**Claims Explanation:**

**Yes, after the required minor modifications.**

The paper studies medical image retrieval across X-ray, ultrasound, endoscopy videos, and CT using image-only self-supervised learning, a shared 4D patchification scheme, and a single ViT encoder. It presents a broad empirical evaluation, including intra-modal retrieval, CT regional retrieval, cross-modal retrieval, scaling analysis, and downstream classification. Overall, the reviewers found the empirical study comprehensive, and the unified encoder design and MRI zero-shot retrieval results are likely to be of interest to researchers working on medical image retrieval, medical self-supervised learning, and medical foundation models.

The main concerns raised by the reviewers are that several claims in the original submission are broader than what the experiments directly support. In particular, the paper should avoid broad representation learning claims beyond retrieval, should not claim uniform SOTA across all datasets and metrics, should qualify the no-customization statement, should describe the cross-modal experiments as anatomy-level rather than pathology-level retrieval, and should present the scaling results as empirical observations rather than a general scaling law.

The authors’ responses address these concerns in a concrete and satisfactory way. The revised manuscript should therefore present the main contribution as a single ViT-B-scale encoder trained with visual-only SSL that achieves strong and competitive zero-shot medical image retrieval across heterogeneous medical imaging modalities. It should also describe Table 3 as anatomy-level cross-modal retrieval based on shared anatomical regions or body parts, and present the scaling results as preliminary empirical trends rather than a universal scaling law.

**With these revisions, the main conclusions are adequately supported by the experiments.** The revised manuscript should incorporate all of the following required changes, which were **raised in the reviews and addressed in the authors’ responses**:

A. Claim Calibration

Revise the paper’s claims according to the narrower framing proposed in the authors’ responses. In particular, adjust the statements about SOTA, representation learning, no-customization, and scaling law to a more calibrated framing: M3Ret is a single ViT-B-scale encoder trained with visual-only SSL that achieves strong and competitive zero-shot medical image retrieval across heterogeneous modalities.

Also reframe Table 3 as anatomy-level cross-modal retrieval. The positive samples are defined by the same anatomical region or body part, rather than the same pathology, finding, disease, or patient. The MRI results should also be described as anatomy-level retrieval, and Qwen2.5 should be noted as being used only for anatomical label mapping during evaluation.

B. Evidence and Comparisons

Add the evidence and comparison updates described in the authors’ responses. Include the 95% bootstrap confidence intervals for Tables 1–3, and describe overlapping confidence intervals or very small margins as comparable rather than clear improvements.

Also add the modality-specific baseline results, including FetalCLIP and EndoViT. Present the results as a specialization-versus-unification comparison: specialized encoders can be strongest in their target modality, while M3Ret provides a single visual-only encoder and retrieval protocol across heterogeneous modalities.

Finally, add the discussion of the CT-X-ray exception where MAE outperforms SimDINO. Present the volumetric-structure versus 2D-projection explanation as a plausible hypothesis rather than a causal conclusion.

C. Reproducibility

Add the implementation and protocol details provided in the authors’ responses. This includes query and database construction, self-match removal, identifier-based de-duplication using patient, study, video, or image identifiers where available, post-retrieval-only label usage, and the limitation that no additional heuristic near-duplicate filtering is applied beyond dataset-provided identifiers.

Also describe the mixed-modality pretraining schedule, including the pooled 300-epoch training, the definition of an epoch, same-modality local batches, mixed-modality global batches, the absence of modality-specific oversampling or target modality probabilities, modality sample counts, and local batch sizes.

Finally, clarify the separation between pretraining and evaluation, and add the missing evaluation details, including fixed lesion-size buckets, deterministic uniform 16-frame endoscopy evaluation sampling, and the rationale for using a temporal/depth patch size of 4.

D. Limitations and Impact

Add the limitations and Broader Impact updates described in the authors’ responses. Clarify that CT volumes are resized or padded to a fixed 64-slice voxel grid without enforcing a common physical voxel spacing or isotropic resampling. As a result, the model does not explicitly encode absolute physical scale, which may affect lesion-size retrieval.

The Broader Impact section should also be revised by removing the statement of “no foreseeable risks.” It should discuss possible clinical over-interpretation, ambiguity in defining positive and negative pairs, the fact that the system should not be used as an autonomous diagnostic tool, the need for external and prospective validation, human-in-the-loop deployment, uncertainty reporting, separation from clinical decision-making, and data de-identification and privacy protection.